# Why Do I Choose an Animal Model or an Alternative Method in Basic and Preclinical Biomedical Research? A Spectrum of Ethically Relevant Reasons and Their Evaluation

**DOI:** 10.3390/ani14040651

**Published:** 2024-02-18

**Authors:** Hannes Kahrass, Ines Pietschmann, Marcel Mertz

**Affiliations:** 1Institute for Ethics, History and Philosophy of Medicine, Hannover Medical School, 30625 Hannover, Germany; mertz.marcel@mh-hannover.de; 2Department for Medical Ethics and History of Medicine, Goettingen University Medical Center, 37073 Goettingen, Germany; ines.pietschmann@med.uni-goettingen.de

**Keywords:** basic and preclinical research, animal research, animal model, non-animal research model, ethical reasoning, bioethics, qualitative research

## Abstract

**Simple Summary:**

Some basic and preclinical biomedical research models require the use of animals. It is not always clear which model is best suited to a project—animals or models based on, e.g., in vitro or in silico methods? This choice is influenced not only by personal beliefs and experience, but also by societal debates. Moreover, people often process information unconsciously. In this study, 13 people involved in relevant areas of research were interviewed. The responses were qualitatively assessed and subjected to an ethical analysis. This paper presents 66 reasons why researchers use animals (27 reasons) or alternative methods (39). Many reasons are tied to the work environment (29) and to scientific standards (22). Such reasons are often pragmatic and can only be influenced by individuals to a limited extent. Other reasons were assigned to personal attitudes (11) and animal welfare (4). Even if few reasons can be rejected outright from an ethical point of view, there are good reasons to give some more weight than others, as an exemplary discussion shows. The study raises awareness of the ethical decision-making process and the underlying reasons that we are often unaware of. This can help to reflect on and justify decisions.

**Abstract:**

Background: Research model selection decisions in basic and preclinical biomedical research have not yet been the subject of an ethical investigation. Therefore, this paper aims, (1) to identify a spectrum of reasons for choosing between animal and alternative research models (e.g., based on in vitro or in silico models) and (2) provides an ethical analysis of the selected reasons. Methods: In total, 13 researchers were interviewed; the interviews were analyzed qualitatively. The ethical analysis was based on the principlism approach and a value judgement model. Results: This paper presents 66 reasons underlying the choice of researchers using animal (27 reasons) or alternative models (39). Most of the reasons were assigned to the work environment (29) and scientific standards (22). Other reasons were assigned to personal attitudes (11) and animal welfare (4). Qualitative relevant normative differences are presented in the ethical analysis. Even if few reasons can be rejected outright from an ethical point of view, there are good reasons to give some more weight than others. Conclusions: The spectrum of reasons and their ethical assessment provide a framework for reflection for researchers who may have to choose between animal models and (investing in) alternatives. This can help to reflect on and ethically justify decisions.

## 1. Introduction

### 1.1. Ethical and Regulatory Background for Using Animals in Basic and Preclinical Research

The question of whether and, if yes, how animal experiments should be carried out in biomedical research (i.e., research that aims to benefit, in the end, human health) has not only been addressed in academic contexts, such as animal ethics, in the last three decades or so. It is also considered to be a controversial social and political issue due to housing conditions, the—sometimes perceived—reduced necessity due to increasing alternative methods such as in vitro or in silico methods, and generally the fact that animals are subjected to stress, pain and harm, or even death [1,2]. Furthermore, in contrast to clinical research, where research is conducted as far as possible with humans for the benefit of other humans, animals are instrumentalized for the benefit of humans. This is only different in the comparatively much smaller research in veterinary medicine, which is intended to benefit animals; however, it can be critically argued that also this animal research is ultimately only carried out in order to further human interests (e.g., reducing economic losses in livestock farming due to diseases, or the treatment of pets). In addition, there are well-known objections about the extent to which animal experiments are even “transferable” to humans and/or how much they can advance scientific knowledge in the biomedical field [3]. Whether and to what extent robust knowledge can be generated from animal experiments, and to what extent this knowledge can also be significant for the development of diagnostics or therapies in humans, are epistemological questions, or questions for philosophy of science (e.g., [4,5]). The term *Therioepistemology* was coined for this a ew years ago [6]. Such questions are often inevitably asked in comparison with the question of the extent to which this knowledge can be achieved equally or even better through alternatives, especially in the case of human-relevant alternatives, e.g., patient-derived cells [7]. Although such epistemological questions can also be addressed independently of ethical considerations, it is also argued that animal experiments which do not have sufficient scientific value fail to be ethically acceptable even if all other ethical requirements are fulfilled; Strech and Dirnagl, for example, propose three principles to safeguard and enhance the scientific value of animal research: robustness, registration, and reporting [8]. In the end, thus, questions related to robustness and value of knowledge gained from animal research or alternative methods almost always take place in an ethically sensitive discourse in which ethical norms such as the obligation to generate “social benefit” with animal experiments—which is only possible through scientifically valid research—and norms aiming at the avoidance, or at least reduction, of animal suffering can be identified in the background.

The accompanying gradual change in scientific, social, and political attitudes towards animal experimentation over the last few decades [9] can also put pressure on biomedical researchers. Researchers conducting animal studies are increasingly subject to an externally imposed regulatory and often related ethical obligation to provide justifiable reasons for the choice of an animal model as part of their *external responsibility* (towards society). Conversely, they are also subject to a scientific obligation as part of their *internal responsibility* (towards the scientific community) to provide reasons why not using an animal experiment and choosing alternative methods is justified. Alternatives roughly refer to all approaches that replace animals or substantially reduce their use in the research context. Some call them ‘new models’ [10], meaning, for example, employing in vitro 2D or 3D cell cultures, in silico methods, and new milestone technology (e.g., CRISPR/Cas, IPSC). To a certain extent, this can also include desk research methods such as systematic reviews if their results lead to a reduction in the number of animal experiments in the future [11].

As long as the research questions are comparable, the strategies and approaches can be different. The project is based on the assumption that the researcher has three basic options: use (also) an animal model, use solely alternatives, or forego research. In normative terms, the decisions made must be based on good (=sufficiently justifiable) reasons, be they ethical, epistemic/scientific, or, as the case may be, merely pragmatic. The many small decisions about how to approach a particular research question, or even just one aspect of it and the reasons for it, will be the focus of the following article.

In this context, ‘making choices’ refers to the many small decisions about how to approach a particular research question, or even just one aspect of it. Countless such choices can be made in research projects, in working groups, and even more so in a researcher’s career. As researchers usually pursue many research questions at the same time or have different experiments running, they often work with animal and non-animal models in parallel [10]. It could be argued, however, that there is no real choice between the three options. Various political debates, especially in the European context, have eventually led to significant regulatory changes. One result is the European Union (EU) Directive 2010/63/EU [12]. However, this has been implemented differently at the national level [13,14]. According to this regulation and subsequent national interpretations, (e.g., in Germany), researchers are legally obliged to choose methods that do not require animals (replace), reduce the number of animals in the experiment (reduce), and improve the conditions of the animals used in the experiment (refine), whenever this is scientifically possible [15]. Although this restricts the freedom of research, it is difficult to argue ethically against this kind of restriction.

### 1.2. Decision-Making and the Role of Ethics

Specific and binding restrictions can be seen especially in the field of toxicology. There are catalogues that define very precisely which test method should be used for a particular test and whether there are validated non-animal alternatives (e.g., OECD Guidelines for the Testing of Chemicals, Section 1, doi:10.1787/20745753) and validated alternative methods via the EU Science Hub “EU Reference Laboratory for alternatives to animal testing (EURL ECVAM) [16]”. Thus, the question of a choice rarely seems to arise; either there is a validated alternative test, in which case it must be chosen, or the animal test is permitted and required (although the search for an alternative may be still desirable).

On the other hand, the research questions and objectives in basic and preclinical research are different and more variable than, for example, testing whether a chemical substance causes measurable skin irritation, as part of a toxicology testing. The aim here is to test hypotheses and develop theories based on a research model, e.g., disease model. These research methods and models must, therefore, fulfil other requirements than in the assessment of safety issues. Researchers in this field have more degrees of freedom than in toxicology.

Thus, it is often not clear in basic and preclinical research whether the use of a particular alternative method is really a valid alternative, i.e., if it leads to comparable results to the animal experiment, allows the testing of the same hypothesis or even maintaining the original research question; the epistemological questions are, therefore, not always answered. The use of an alternative can, therefore, often be a kind of experiment in itself. Decision-making generally involves rational considerations, emotions, and interests, both consciously and unconsciously. These considerations have to be balanced against each other. A number of value judgments play a role in this weighing process: *What value do I give to one argument or another? What is a “good” decision?* Ideally, all these relevant aspects are systematically identified and weighed against each other in a structured process based on predefined criteria. Decisions are then systematically derived and fully justified. Given the many small decisions this project has in mind, it is clear that this ideal is more aspirational than achievable. However, from an ethical and scientific perspective, it is desirable to make important decisions, such as the choice of a research model, *as consciously and systematically as possible* [17]. In this context, ethics, as the systematic exploration of values, norms, and principles, and as the critical examination of arguments involving (moral) value judgments, provides a relevant theoretical and methodological background for analyzing and evaluating decision-making, which does not preclude other relevant approaches, such as those from (therio)epistemology or cognitive psychology. In this way, empirically stated reasons that play a role in decision-making can be classified thematically, on the one hand, and ethically evaluated, on the other, following methodological approaches of empirical bioethics [18,19,20], which, in the meantime, have also found some counterparts in animal ethics [21,22].

### 1.3. Aims and Research Context

The aim of this paper is, therefore, (1) to identify a spectrum of reasons for choosing between animal models and alternative methods, empirically based on a qualitative interview study with researchers working in basic and preclinical biomedical research. Individual reasons and categories of reasons are then (2) discussed (exemplarily) from an ethical point of view, including the value judgments involved. The spectrum of reasons, the associated categorization, and ethical assessment provide a framework for reflection for researchers who may have to choose between animal models and alternative methods.

That a certain reflective framework for such decisions might be useful became clear in the context of a larger research network, to which the research presented here also belonged. The R2N—‘Replace’ and ‘Reduce’—consortium was funded by the Ministry of Science and Culture of the Federal State of Lower Saxony (Germany) and consisted of 14 projects (12 life sciences projects, 2 ELSI—ethical, legal, and social issues—projects). The aim was to develop new alternative methods to reduce the number of animals used in experiments or replace specific animal experiments altogether, mainly in basic and preclinical research (https://r2n.eu/, accessed on 1 December 2023). Selected results from one of the ELSI-projects are presented here; related preliminary theoretical work has already been published in [23]. Work from the second ELSI-project that focused more on a perspective from philosophy of science has also been published [4,24].

## 2. Materials and Methods

The results presented, i.e., the spectrum of reasons and the subsequent ethical discussion, are based on empirical and theoretical work [23].

### 2.1. Interview Study

An in-depth literature review and preliminary interviews with various experts concerned with animal experimentation and/or alternative methods were conducted to become more familiar with the research field of basic and preclinical research involving animal experiments, and to develop the interview guide for the qualitative interview study. The conduct of the interview study was deemed unobjectionable by the ethics committee of Hannover Medical School.

*Recruiting:* A snowball recruitment was initiated through expert recommendations via the R2N consortium; however, it was deliberately decided not to recruit researchers from the R2N consortium itself. The contact was successful in most cases because we were able to establish a known connection to researchers in the requests. We asked the interviewees about other researchers who might be open to a possible interview.

The ‘hard’ inclusion criteria were: (1) experience with animal models and/or non-animal models in basic and preclinical research; (2) the possibility to conduct the interview in English or German; and (3) consent to participate and to the publication of results in an anonymized form. Further categories applied to define the sample were different career levels (PhD to senior researcher), affiliation, and gender. The end of recruitment was determined by the saturation of content that emerged from the successive analysis of the interviews. Rather than aiming for representativeness in the sense of quantitative research, qualitative research aims for diversity, depth, and range of topics. Thus, the significance of the results in qualitative research comes from their contentual comprehensiveness, not from numbers (see Section 4).

*Interview Organization*: The interviews were carried out as episodic interviews [25], a combination of a guideline-based/semi-structured interview with a predetermined order of (open-ended) questions and a narrative interview that aimed to incite narratives on the experience of deciding between animal testing and alternative methods. The interview guide was divided into three parts. The first focused on the individual understanding of what alternative methods are. The second concentrated on specific decision situations, and the third centred on the research environment and other contextual factors. The focus of the interviews was on parts 2 and 3.

The interview guide was pretested with bio-scientists of the R2N consortium, as they were similar to the desired interviewees in terms of education and experience. The interview guide was slightly revised in terms of wording. The interviews were conducted by telephone in German and recorded with prior consent (primarily by IP, at the first interviews together with MM). The interviews were subsequently transcribed and the resulting texts were analyzed qualitatively (see *Analysis and categorization*).

*Analysis and categorization*: Qualitative content analysis was used to analyze the texts, mainly following the method of Mayring [26]. It is an interpretative analysis method from social science research for processing qualitative data, based on the premise that the analysis starts from everyday processes of understanding and interpreting linguistic material. The method is, thus, based on psychological and linguistic theories of everyday text comprehension.

At the beginning, the interview transcripts were carefully read line by line, marking passages entailing value judgments and the associated argumentation, using the MAXQDA 2018.2 software. The content was then characterized using existing concept (deductive) and text-driven/generated (inductive) codes or ‘headings’ (e.g., time commitment, trust in alternatives, or career with a subheading on pressure from superiors but also prestige). The concept-driven elements were based on a certain understanding of value judgements and their structure developed within the project, the so-called Value Judgment Model (see below). This is necessary because in the empirical analysis step, value judgments are theoretical constructs, i.e., they cannot be directly empirically ascertained. In order for value judgements and their components and justifications (e.g., in the form of descriptive assumptions, but also values, interests or emotions, etc.) to become recognizable (“visible”) in a text, an “operationalization”, here in the form of a conceptual model, must, therefore, be used.

The text-driven code generation was successively reconciled with the more theoretical-driven coding. Once the content of all interviews was coded, the codes were compared, and similar codes were merged into paraphrases that reflect the core of a stated reason for a decision. These paraphrased reasons were thus a condensed reproduction of the statements from the interviews, which was accompanied by a certain degree of abstraction. This allows concise reasons to be extracted from the interviews, based on multiple passages from multiple people, making them less subjective. The analysis was conducted step-wise by one author (IP), with interim results discussed repeatedly within the team (HK, MM), so that the code tree evolved continuously. In a further step, the paraphrased reasons were categorized into reasons for using alternative models and reasons for using animal models. Each was further divided into four categories, mainly based on inductive categorizing: personal attitudes, work environment, animal, and scientific. One author (HK) took the lead on this, with ongoing review by a second author (IP).

### 2.2. Theoretical/Ethical Analysis and Evaluation

*Value Judgment Model*: The reasons for choosing an animal model or an alternative method always involve value judgements; they are an inherent part of the logical reasoning structure. Therefore, a Value Judgment Model (VJM), was developed beforehand as part of the project [23], and was applied as part of the coding, identifying relevant content (see *Analysis and categorization*), and as part of the subsequent analysis. In brief, according to that model, value judgments are the logical conclusion of an argument that consists of at least one value-sensitive descriptive premise and at least one evaluative premise, see Table 1.

Due to the subject matter, the value judgments themselves always boiled down to the fact that an alternative method was, to put it simply, better or worse than the animal experiment with regard to certain aspects, and vice versa. These value judgments were not always explicitly stated, but regularly arose from the context of the conversation. The premises of the value judgments were thus often more interesting. Explicating the evaluative premises or assumptions, which were often only implicitly provided when scientists gave reasons for their choices, serves to clarify and confirm the relevant values that are involved. Descriptive premises or assumptions are characterized by the fact that they, in principle, can be empirically verified.

Both premises could be supported by further descriptive or evaluative/normative premises which function as their backings (e.g., referrals to epistemic processes, infrastructure, work environment, interest(s), emotions, desires, needs, and further or “higher” values or ethical principles). These considerations are less present or conscious, but can influence reasoning and, thus, decision-making processes, see Table 1.

By applying the VJM, non-explicit aspects of reasoning can be logically reconstructed and analyzed in detail. When applied to the interviews, the value judgments themselves or the descriptive and evaluative premises not explicitly mentioned had to be reconstructed either hermeneutically or rationally. Hermeneutically reconstructed means that the most probable statement containing such a premise was worked out on the basis of further statements in the interview or by interpreting a statement in the context of other statements. Rationally reconstructed means that, using the principle of charity (“charitable interpretation”), the missing premises are supplemented in the way that a rational actor would formulate them in order to provide the most plausible (“best”) justification.

Table 1 (slightly modified from Table 1 in [23]) lists three exemplary value judgements and illustrates the evaluative and descriptive premises to each. In each example, one formulation was taken from the interviews and the other four were reconstructed by the authors. For reasons of space and in order to keep the analysis presented in the paper more focused on the reasons for choosing alternatives or animal models itself, not always will all premises of the argument be mentioned later in the examples.

*Ethical framework for animal research ethics*: The VJM makes it possible to (better) identify value judgments in texts where reasons are given, and reveal (or reconstruct) their justificatory role when such reasons are given. However, the model is not intended to evaluate reasons from an ethical perspective. This requires an ethical framework.

In order to evaluate and reflect on the reasons identified (HK and MM), an established so-called principlism approach [27] to bioethics was used. Such approaches use mid-level ethical principles (abstract norms) to identify, analyze, and ultimately evaluate actions. However, these principles are not in a predefined hierarchical order; which principle has the most weight in a particular case, and the duty to be followed, depends on the context and particularities of that case, as well as on the concrete analysis and subsequent argumentation.

Different principlism approaches have been developed for different areas of bioethics. A principlism approach for animal research ethics has been proposed by DeGrazia and Beauchamp [28]. According to this approach, the suffering of laboratory animals can only be justified if it is absolutely necessary (*Principle of No Alternative Method*), experiments may only be as severe as necessary (*Principle of No Unnecessary Harm*), and harm done to animals cannot be unlimited (*Principle of Upper Limits to Harm*). Moreover, basic care must be guaranteed at all times, which also means during the actual experiment (*Principle of Basic Needs*). Furthermore, it is to be expected that benefits (for humans) and harms (for animals) have to be weighed against each other, both by the researchers themselves and the respective responsible commissions; it is required that the benefit is sufficiently high to be able to justify any harm to animals at all (*Principle of Sufficient Value to Justify Harm* and *Principle of Expected Net Benefit*). The approach explicitly does not aim to replace the established “3Rs” (replacement, reduction, and refinement [29]), “but to add complementary content for animal research ethics that the 3 Rs framework fails to provide” [28] (p. 310). Even if there are other approaches to animal research ethics, we considered this approach to be suitable for addressing essential ethical aspects of animal experiments. In addition to this approach, the importance of maintaining/upholding scientific validity in animal experiments [8,29], as well as scientific integrity [30], have been emphasized as additional ethical principles.

## 3. Results

We contacted 30 potential interviewees (including follow-up) and conducted 13 interviews between March and June 2020 (4 people declined the request and 13 did not respond to our invitation). Further characteristics of the interviewees can be found in Table 2.

In the following, we will give an overview of the results of the interviews, firstly of the understanding of alternative methods, and secondly, regarding the spectrum of all the reasons identified for the use of animal or alternative models. Additionally, some of the reasons were evaluated ethically. Thereby, we will only take up a part of the reasons identified to illustrate relevant ethical aspects.

### 3.1. What Is Understood by “Alternative”?

The interviews indicated that there does not seem to be a consistent definition of the term “alternatives”. What researchers understand by “alternatives” was answered very differently. 

Some defined it by means of concrete methods: “So, a cell culture experiment would be an alternative for me” (original statement from an interview transcript: Interview 9). There is often a very specific understanding of what an alternative is. This leads to the fact that the perspectives refer to very specific contexts and are only transferable to a limited extent. Some researchers defined alternatives with the help of the 3Rs concept: “For me, alternative methods are methods that do not require animal material and completely replace an animal experiment or part of an animal experiment” (Interview 8). In this case, “alternative” is also understood as a completely animal-free alternative, where animal products (e.g., blood, enzymes) are no longer required, for example, as a nutrient solution. However, this understanding was not necessarily shared when referring to the 3Rs, as which of the latter was used for the understanding was different: “Well, I would understand alternative methods to be techniques in which, on the one hand, the stress that occurs within animal experiments/to which the animals are exposed can be reduced. And, on the other hand, animal experiments can be totally replaced as a whole” (Interview 1).

Others referred to the definition of an animal experiment from the German Animal Welfare Act [15] and derived their definition of alternatives from it: “For me, animal testing alternatives would actually mean that no animal is used in the sense of an animal test” (Interview 3). However, it is clear that there does not seem to be a clear boundary as to what is considered an alternative and what is not. Thus, in the discussion about animal experiments and alternatives, it is worthwhile determining more precisely what is actually being talked about, especially when the discussion is not only conducted within the research community but also with a broader public.

### 3.2. What Is the Spectrum of Reasons?

There were 846 passages coded from which 66 specific considerations related to the choice of a research model were derived. “Reasons” in the following can refer directly to a value judgment. However, this is rather rare, since, as stated above, the value judgments were not always made explicitly, but usually resulted from the context of the conversation and, put simply, can be broken down to “Alternative is better than animal experimentation” or “Animal experimentation is better than alternative” in the respective case. Therefore, “reasons” usually refer to descriptive or evaluative premises that justify such a value judgment.

Numerous reasons were given for both the choice of the alternative (39; 59%) and the animal model (27; 41%). In both sub-spectra, the reasons were distributed across four areas, with the Working Environment (WE) accounting for the largest share (45/37%), closely followed by Science (S) (29/37%). The areas Personal Attitudes (PA) (12/22%) and Animal (A) (7/4%) were quantitatively smaller. All 66 reasons identified are listed in Table 3.

Additional subcategories were introduced for Working Environment (reasons concerning work organization; research climate within institution; expert opinion and research funding; peer-group/Scientific community; education and teaching; society; technical development) and for Science (research questions and approaches; results; translation/usability; publications) because these two areas cover a wide range of topics.

The qualitative spectrum indicates what reasons can, in fact, play a role, at least in the reconstruction and regarding possibly subsequent questions of justification. Nevertheless, they are not effective everywhere and all the time, or endorsed by all researchers. It does, however, provide an overview that can be used to systematically select significant aspects from an ethical perspective.

### 3.3. What Are Exemplary Ethical Dimensions of the Reasons?

*Personal attitudes*: In the category of personal attitudes, there was, not surprisingly, a certain range of reasons. Not all reasons in this category refer only to internal or subjective criteria—as is the case when, for example, emotions play a stronger role (evaluative premise): “I choose the alternatives because experiments with animals are stressing me emotionally/psychologically” (PA1.3, Table 3). For instance, the reason (evaluative premise) “I choose the animal model because it can be justified and is ethically acceptable” (PA2.1) also refers to an external basis for justification, even if it is not further explicated in the reason itself. The evaluative character of this reasoning becomes clear, as the reference that animal experimentation is justified and ethically permitted could be based on the current societal norm setting, which basically allows animal experiments. One could also refer to the defined procedure for third party approval (competent authorities), which can be regarded as safeguards for ethically defensible research involving animals. However, the reason could also refer to scientific and ethical arguments that are to be understood independently of or in addition to an examination by an animal ethics committee, but which initially elude further intersubjective examination due to a lack of explication. It is, as an unspecified reason, initially only an assertion that the animal experiment is justified, without reference to ethical principles. So, personal attitudes can be grounded in (a) values/principles (“protecting a human being from ineffective or harmful drugs is a higher value than refraining from animal experiments,” PA2.4), (b) emotions (PA1.3 see above), or (c) interests (“I am curious to try new things,”, PA1.1), which are all evaluative premises.

Ethical dimensions: None of these personal attitudes was ‘prima facie’ ethically better or worse. However, the background of the evaluative premises can be further explored. The reason of being curious (PA1.1), for example, would not be ethically sufficient as the sole reason for refraining from animal experimentation in view of the *Principle of Expected Net Benefit*, if the use of an alternative would reduce the expected social benefit. Furthermore, a general attitude to ‘delegate’ the decision to a ‘higher’ level of decision-making could be problematized, as there is then no ethical reflection and justification of one’s own. The reasons discussed also contain descriptive statements that can be analyzed. Following the VJM, the descriptive premise “the established approval procedure is most likely to lead to a ‘correct’ decision about moral permissibility” could be rationally reconstructed from the interviews, as this makes the delegation of ethical assessment to, e.g., animal ethics committees plausible. In the case of independent reasons, however, the “justified” could also refer to scientific or epistemological reasons related to the fulfilment of the norm of scientific validity.

Personal attitudes are thus often expressions of different implicit assumptions. In PA1.3, an implicit premise such as “my active role and the suffering I am directly confronted with affect me too much personally” may play a role. This also implies an evaluative backing that suffering (in general or specifically also regarding animals) is ethically not good—otherwise it would not be experienced as emotionally burdensome.

Generally spoken, personal attitudes are legitimate for any person. They cannot serve well as impersonal normative justifications, claiming to give intersubjective comprehensible reasons for the rightness or wrongness of an action. However, they can serve as personal justifications, claiming only to state why this particular person has judged a particular action to be right or wrong and/or has preferred it or refrained from it (cf. [31]). This is because, for example, emotions or desires are not generalizable, nor are personal experiences or personality structures and associated interests. However, some issues should still be discussed from an ethical perspective. These considerations are referred to as *secondary interests* in the concept of conflicts of interest: interests that are not directly related to professional activities as a researcher (e.g., personal curiosity of a scientist, completing a doctoral thesis as part of pursuing a career, or just earning a living). It is important to note that primary and secondary interests are often not in conflict with each other, but can even have positive effects. However, if they are conflicting with *primary interests* (e.g., the pursuit to produce relevant and valid findings, disseminating results in the scientific community, or adherence to the six principles of animal research ethics), they should not influence the professional judgement inappropriately [32]. This could violate a general norm of objectivity or disinterestedness as expressed, for example, in the Mertonian norms/ethos of science [33]. In addition, if someone argues that conducting animal experiments stresses them emotionally, a phenomenon that is now well known and is treated under terms such as “compassion fatigue”, “mental stress”, or “moral stress” (e.g., [34]), this probably (implicitly) refers to a general norm of non-maleficence, such as “do not harm others” or “thou shalt not kill”. This would, to a certain extent, also fulfil the *Principle of No Unnecessary Harm*. Such norms are presumably shared by significantly more people than the motivation of “striving for media attention,” which would primarily satisfy a personal need.

In sum, personal attitudes influence decision-making, such as on a research model, or our positions towards certain professional issues (e.g., conflict of interest issues or scientific integrity). When considering esp. evaluative premises, it is necessary in each case to examine whether emotions (but perhaps also interests) are based on general and consensual moral norms that appear as implicit evaluative backings. In such cases, however, the norms should be brought into focus because they allow for an impersonal normative justification. Therefore, it is ethically important to look at the further justifications of such seemingly purely personal attitudes.

*Work environment*: Some scientists argue that “alternatives are associated with smaller amounts of lengthy bureaucracy (e.g., no approval procedure)” (WE1.2). Ethical dimensions: The evaluative assumption behind this might be that “lengthy bureaucratic processes are bad for/are a hindrance to research”. It should be noted that this is a specific perspective that emphasizes efficiency, which can be understood as a shared value or (ethical) principle in science, and especially in research within the healthcare system, given that public resources are being spent. However, in the research context, scientific validity or ethical integrity is probably more important than efficiency, i.e., a higher value must be placed on validity or integrity when comparing research models. This does not mean that efficiency cannot be taken into account, but decisions should not be made only in favor of efficiency; choosing an alternative method only because it means less bureaucratic effort would be ethically questionable when scientifically, an animal model would be preferable, as this also may violate the *Principle of Expected Net Benefit* (=social benefit is only possible when the research model is meaningful for the research question and the results robust). Additionally, the review system for animal experiments, which may be perceived as burdensome bureaucracy, is not an end in itself, but is, in a certain way, an operationalization of the *Principle of Sufficient Value to Justify Harm* and the *Principle of No Unnecessary Harm*.

Other aspects of the work environment can also play a role in the choice of a research model. Some researchers, for example, report that “superiors are demanding or are explicitly supporting the use of alternative methods” (WE1.9) or “my supervisor has decided so” (WE2.4). Ethical dimensions: There is a mutual dependence or dependent relationship in academia (as well as in private sector). It is important to be aware of these and their possible influence on our decisions via the definition or modification of descriptive assumptions or, above all, evaluative assumptions (and the values behind them). This is important because conflicts can arise between one’s own evaluative assumptions and those decisions and actions that one must follow or support as a team member. In extreme cases, this can lead to moral distress, a feeling of helplessness that what one feels to be ethically right is not being done, and that one cannot (adequately) influence this due to hierarchies.

A further important context factor that was brought up in the spectrum of reasons is the relevance of available or necessary infrastructure (WE1.4 and WE2.2–3). If animal laboratories and related expertise are already available, and the switch to an alternative method is, therefore, assumed to be too costly, this may influence the decision. Conversely, researchers who only work in vitro, for example, will have neither the infrastructure nor expertise for animal research, and if they do want to resort to animal experiments, they will have to outsource them (e.g., via cooperation partners) (WE1.5, Interview 10). Ethical dimensions: Similar to bureaucratic requirements, actions based on existing or required infrastructure are not, per se, unethical. Again, however, it would become ethically problematic if scientific validity or integrity were negatively influenced, or, in extreme cases, if animal experiments were carried out that were, strictly speaking, not absolutely necessary (*Principle of No Alternative Method*)—simply because the infrastructure would only allow animal experiments to be performed.

We then extracted the argument that some researchers continue to work in basic and preclinical research with animal models because “science has 50 to 60 years of experience with certain animal models” (WE2.6). Ethical dimensions: The plausible descriptive premise is that there is a lot of experience and achievement with these animal models that have helped to advance science. Whether these benefits exist and, if yes, what their magnitude is, has to be addressed very field-specifically and should be documented in a comprehensible and systematic manner. If this is based more on financial considerations (e.g., avoiding the need to invest in infrastructure), this is more problematic from an ethical point of view, probably violating the *Principle of No Alternative Method* and *Principle of No Unnecessary Harm.* In specific fields and to a certain extent, there may be legitimate reasons why a more suitable model cannot be implemented in a specific situation, i.e., does not represent the “best choice”. However, scientific factors and animal welfare should usually be decisive.

*Science*: Reasons belonging to this part of the spectrum are based on statements about specific fields of research. This means that for evaluative statements such as “I find the transferability of animal models to humans inadequate” (S1.5) or “I lack confidence in the potential of alternatives” (A2.5), a “related to my research field XY” should be added in the mind. This contextualization becomes clear in statements such as “complex questions/interactions can only be investigated in the whole organism” (S2.2), which seems to be more obvious in some fields and for some research questions than others. Ethical dimensions: Such an assumption forms, more or less directly, the descriptive premise in the justification of an animal experiment, likely to be combined with an evaluative premise based on the *Principle of No Alternative Method*. In the theory of principlism, it is also conceivable that the *Principle of No Alternative Method* is *specified* accordingly on the basis of this descriptive information, so that it says, for example: “If no alternative methods are available or suitable for studying complex issues or interactions in the whole organism, then, prima facie, an animal experiment may be carried out”.

Other researchers argue that (regarding their research field): “I get publishable results faster when I am using an alternative” (S1.12). Ethical dimensions: This reason is similar to the earlier example, in which an alternative research model is only chosen because of the possible lower bureaucracy, although animal models would perhaps make more scientific sense and thus make a higher social benefit possible (*Principle of Expected Net Benefit*). In the example here, a problematic influence by secondary interests (number of publications as a career benchmark) is perhaps even more conceivable. However, if it is assumed that the results can be used in a comparable way by applying the alternative methods, the tide can turn. Only published findings can fulfil the central promise of generating value through research, as only what is published can be taken up scientifically and, at best, later translated into healthcare. If usable results can be produced and published more quickly by using alternative methods, the value of animal experiments for a comparable question is reduced, as this also calls into question the fulfilment of the *Principle of Sufficient Value to Justify Harm*: The sufficient expected social benefit to inflict harm on animals is reduced if sufficient value can be generated via alternative methods, and then even faster.

In contrast to such examples of reasons that can be attributed more or less directly to justifications using ethical principles, there are pragmatic considerations, for example, that “there is suitable infrastructure on site” (WE 1.4), that something could be ”realized more quickly” (WE 1.3) or that “there are also legal requirements” (S2.9) Ethical dimensions: The different dimensions of use or benefit should be considered here. For research to be of benefit to society, the results must be published in full and in a timely manner. There is also the legitimate interest of the individual researcher in commercialization by industry (“industry often demands results from animal models, otherwise it is hardly possible to commercialise our results”, S2.6). Even if commercialization appears to be necessary in the current scientific system to continue research through better funding, the *Principle of Expected Net Benefit* should be carefully respected. So, such secondary interest should be given less weight in the event of a conflict, as the primary interest should be the aim of rapid and comprehensive publication.

Another issue that should be discussed is that both animal models and alternatives “imitate the human organism only to a limited extent” (S1.1+5 and S2.7). We should be clear what kind and scale of uncertainties [6] we want to accept when going from preclinical to clinical research and, thus, begin to involve humans. Do these uncertainties tend to be larger, equal, or smaller for alternative models? Perhaps uncertainties related to the use of alternative methods impress more than the common and known uncertainties in animal models. Finding a clear answer to this question is not made any easier by the replication crisis (e.g., [35,36]) and biases in the risk assessment based on investigator brochures [37,38]. Ethical dimensions: From an ethical point of view, it concerns the meaningfulness and the (social) benefit, which must be empirically proven and critically reflected on a field-specific basis (*Principle of Expected Net Benefit* and *Principle of No Unnecessary Harm)*. Against the background of dynamic technological developments (e.g., new opportunities through artificial intelligence), this should be performed on an ongoing basis.

Researchers have to continually find sources of funding and, thereby, orient themselves to the external (public) research funders. Investments in promising in vitro or in silico methods (such as organoids, organ-on-chip, human-cell-based models, or computer simulations), or more and better systematic reviews as desk research, as well as the qualification of personnel to use them, is a strong driver for a shift more towards alternative methods, and if the investment costs are funded, this is beneficial (WE1.10+13). Ethical dimensions: Ethically, it is rather trivial that if funding is available for the use or development of alternatives, the researchers are competent to use or develop them and it can be assumed that comparable research questions with comparable (expected) social benefit can be addressed with them, the alternatives are preferable to animal experiments. If a viable alternative is obviously available or can probably be developed, an animal experiment would violate the *Principle of No Alternative*, which is quite uncontroversial and also, as mentioned in the introduction, legally defended. It is, therefore, more interesting to ask what happens if the funding for alternatives is not available or is insufficient, even though it would be conceivable to use or, especially, develop a viable alternative. As the resources are allocated in different funding lines, it may be the case that no funding is available for the development of infrastructure and the qualification of staff for research with alternative methods, but funding is available for the acquisition of a certain number of mice. Formally, an evaluative premise such as “It is good/right to conduct an animal experiment (at least better than not conducting any research) if an alternative is generally available but cannot be adequately funded in this case” is able to provide a justification. However, accepting such a premise obviously opens up countless exceptions to the *Principle of No Alternative Method.* Thus, if alternatives would be available or could conceivably be developed, insufficient funding is not an ethically convincing argument for the use of animal models. Still, with reference to the so-called ought-implies-can principle—simplified: one can only demand normatively what is also realistically realizable (e.g., [39]), it could be argued that animals are the only possible research model in this case. Nevertheless, following the *Principle of No Alternative Method*, one could still argue that the conclusion in these cases should rather be “no animal experiment if an alternative is available but financially unfeasible”—even though this must be examined in each individual case (e.g., to what extent is the alternative available but not feasible, to what extent would it be suitable).

*Animal welfare:* All reasons that are subsumed under this category in the spectrum of reasons are either easily complemented by principles of animal research ethics when the reasons are referring to descriptive assumptions, (e.g., “I want to avoid animal suffering”, A1.1) or are, in the end, specifications of these principles (e.g., “thereby allowing various pre-experiments that may otherwise cause animal experiments” (A1.2), *Principle of No Unnecessary Harm)*. Given these principles, current animal protection laws, and social and political movements that aim to improve animal welfare or even rights, such reasons should be guiding considerations in decisions. Ethical dimensions: Reasons such as “I want to avoid animal suffering” (A1.1) are very straightforward ethically, as they can be directly subsumed under the *Principle of No Unnecessary Harm* or to more general ethical norms that are oriented towards non-maleficence. If this kind of reason is understood in absolute terms, it inevitably leads to an abolitionist position, i.e., the complete abolition of animal experiments; this would go beyond the *Principle of No Unnecessary Harm*. However, if it is understood in relative terms, it leads to the well-known question of weighing up what kind and what degree of harm is permissible (“necessary”) in view of a possible benefit, which is represented in the *Principle of Sufficient Value to Justify Harm*. Defined constraints for weighing are also conceivable in favor of scientific considerations (e.g., validity) and their related ethical values (e.g., expected social benefit). Still, the appropriateness of the weighing should always be checked in a review process by third parties to mitigate possible biasing effects of secondary interests (see *Personal Attitudes*).

Other interviewees pointed out the consequences of failure of the experiments and compared this between animal experiments and alternative experiments: “The consequences of a failed experiment are much less critical than in animal experiments” (A1.3). Ethical dimensions: This reason, besides following in a way the *Principle of No Unnecessary Harm*, as unnecessary harm would be the case when the animal experiments fail, seems also to refer to a form of the precautionary principle (“err on the side of caution”): It is better to lose a potential social benefit that could have been generated by animal experiments than to cause possible unnecessary harm to animals. When balancing the principles, the *Principle of Expected Net Benefit* is, therefore, given less weight than the *Principle of No Unnecessary Harm*. 

The reason A2.1 for conducting animal experiments is somewhat more complex: animal experiments “[…] are necessary and before someone does it who doesn’t care about animals, I prefer to do it myself”. Here, “objective” assessments—the experiment is scientifically necessary—are combined with “subjective” attitudes. Ethical dimensions: On the one hand, it is clear that the *Principle of No Alternative Method* must be fulfilled; otherwise, the animal experiment would not be necessary. On the other hand, however, the interviewee has doubts that the *Principle of No Unnecessary Harm* and perhaps also the *Principle of Upper Limits to Harm* are observed by all animal researchers, which is a concern for him/her. This means that even if he/she might prefer not to carry out animal experiments, he/she will still carry them out to ensure that at least in these animal experiments the principles are met.

Finally, when considering the reasons for the *Work Environment* and *Personal Attitudes* categories, it is worth reflecting on the extent to which it is permissible to restrict animal welfare in favour of mere pragmatic contextual factors (e.g., infrastructure or funding). Ethical dimensions: When an ethical stance is taken that always gives priority to the moral point of view or the moral position over other points of view, especially over self-interest (e.g., [40,41]), the answer is relatively clear: it is never permissible, although perhaps not always avoidable in reality. In any case, it then only seems to be defensible at all if the pragmatic reasons can (indirectly) refer back to ethical values or principles, e.g., to the *Principle of Expected Net Benefit*. In a sense, the argument would then be that not conducting animal experiments or conducting them in a more limited way would lead to fewer expected social benefits, which is why at least some pragmatic reasons in a non-ideal world must be regarded as justification. However, the extent to which such an argument can be used in individual cases must be subject to critical discussion.

## 4. Discussion

We empirically identified a spectrum of 66 reasons for choosing an alternative method or an animal experiment in basic and preclinical research. A considerable number of reasons can be found in the categories belonging to work environment (>37%) and the scientific context (>29%). Although this does not imply a conclusion about the legitimacy or relevance of the arguments, it is noteworthy that such reasons receive less attention in the academic ethical debate, which rather tends to focus on issues of animal or research ethics (e.g., animal welfare or harm–benefit analysis [17,42,43]. Interestingly, such reasons tend to be addressed more in the public/societal debate (e.g., [9]), both in the defense of animal experiments (here mainly those that refer to the scientific context) and in the criticism of them (here rather, although by no means exclusively, those that refer to the work environment; e.g., to argue that it is tradition, existing investments in infrastructure, and hierarchies that lead to animal experiments, not “ethical” reasons). Thus, while some academic ethicists may, therefore, argue that only principles that deal with the animal and the necessity of its use in a laboratory are allowed in ethical decision-making, it seems important to understand reality and accept the actors’ scope of action and motivations as an empirical fact. Nevertheless, it is not surprising that some reasons in the spectrum can also be more or less directly associated with basic animal and research ethics principles (e.g., protection of non-human and/or human animals). In this context, however, more general values and principles were also reconstructed as a normative reference (e.g., “thou shalt not kill”). In addition to the value-based reasons, others were more pragmatic and argued with, for example, infrastructure or funding. From an ethical perspective, they may not be equivalent, but, as others have shown [10], they are hard constraints in reality and must, therefore, be considered. Moreover, personal attitudes were identified as relevant for the decision-making, albeit not necessarily for impersonal (intersubjective) normative justification. For the latter, it is important that personal attitudes are congruent with commonly agreed values and that effective measures are taken to promote professional ethical conduct. It must also be acknowledged that there are (legitimate) secondary interests that can influence professional judgments. The personal (secondary) interest, for example, not to attract attention (“society demands more alternatives or animal experiments are criticized more than before,” WE1.17) versus the professional (primary) interest to generate relevant valid findings (“the modes of action can only be studied in the intact organism or in the whole organism,” S2.2). As it is difficult to identify illegitimate influences on professional judgment (blind spots), however, it may be helpful to think systematically about particular reasons or considerations. We hypothesize that the pragmatic reasons, which might often have an influence on the decision, are not reflected so much ethically because they are not perceived as “ethical” or “ethically relevant” (since they do not directly violate ethical principles). Only the clarification that these reasons may indirectly correspond to ethical principles or contradict them would also enable a more reflected decision-making.

There is a generation of researchers who have worked without new milestone technologies (e.g., artificial intelligence) for many years and have based their academic careers on animal models. It would be wrong to imply that this generation is more critical of new milestone technology overall, but this impression was described for individual cases in the interviews (“Reviewers tend to come from a generation in which animal experiments are recognized above all,” WE2.5). It is perhaps not yet imaginable to be able to understand complex things in a life form without being able to grasp them in a living organism. Virtual and organic replicas of reality or information-processing computer systems, however, make ever-greater demands on the justification of the necessity of particular animal models. In this discussion, fundamental value questions also play a role: What are meaningful outcome dimensions for making statements about effects in humans, what implications arise from the interpretation of the data, and how do you deal with uncertainty and risk?

As humans generally, we are often not fully aware of all the considerations that lead to our decisions. Moreover, it is hardly possible to describe the full process of consideration within an explanatory statement. By paraphrasing and reconstructing, it was possible to create a spectrum of 66 distinct reasons, each based on (several) statements from the interviews. Thus, by acknowledging that decision-making in specific research areas refers to individual cases, specific decisions are not reproduced one-to-one in this paper. Nevertheless, this spectrum provides a framework for reflection on individual behaviour and may help to make one’s own perspective in the ethically tense field of basic and preclinical research more conscious and stronger. The analysis has shown that it is important and valuable to distinguish between descriptive and evaluative premises and other elements of justification (backings). Within the descriptive and evaluative premises, it is again useful to distinguish whether they are statements about, for example, infrastructure, financing, harms or benefits, or statements expressing things such as emotions, interests, or needs. While one can use the VJM analytically for this purpose, in practice it will probably suffice to make it clearer that there will always be at least one descriptive and one evaluative premise justifying the value judgment in relation to a research model. Being clear about this, and especially trying to make the evaluative premises more explicit, can help to check whether the value judgment is sufficiently supported. Because if it is assumed that a value judgment is the conclusion of an argument, it is not sufficient to merely demonstrate the truth of the descriptive premises. It must also be shown why the evaluative premises are justified, e.g., by referring back to established or at least intersubjective defensible ethical principles or values. The tendency in the natural sciences may often be to secure the descriptive side as evidence-based as possible, but to forget which value-related premises are already implicitly assumed in order to make a value judgment.

We faced several challenges in the recruitment phase. A first attempt, to limit and define the scope of the decisions-making-situation, focused on Alzheimer’s research. We contacted a large number of researchers in this field multiple times by mail and telephone. We only received one response. One possible reason for the failed recruiting could have been the COVID-19 pandemic. The interviews were planned and conduced in the period from April to June 2020 (during “lockdown” in Germany). It is possible that this circumstance led to a lack of availability. However, we were unable to verify this hypothesis. We then decided to change our strategy and no longer limit ourselves to Alzheimer’s research.

The theoretical saturation in this project was reached; nevertheless, it is important to note that we might have missed reasons, for example, by using of different ethical theories or approaches. Regarding the latter, we did not analyzed or discuss (therio)epistemological issues, and have decidedly not taken a perspective from the philosophy of science, as one of our colleagues in R2N did [4]. Our focus was on the ethical value judgments and associated reasons. However, we have implicitly addressed epistemological reasons insofar as they are presupposed in certain ethical norms. For example, in norms that demand scientific validity; but basically already in the *Principle of No Alternatives*, which presupposes the answer to the epistemological questions of what knowledge can be generated by animal experiments or alternative methods and how “good” this knowledge is. Furthermore, other reasons might have been found in a sample with more female respondents. In our sample, senior male researchers in Germany where the most dominant group. The spectrum presented can and should, therefore, be complemented by further qualitative research. However, it is important to notice that qualitative research strives for diversity and range of topics and not representativeness in a statistical sense. On a personal level, not all 66 reasons are likely to be equally important, and some will not resonate at all. Our qualitative approach does not allow us to say anything about the frequency of the reasons given. It is possible that some reasons are primarily due to particularities such as the bureaucratic effort and associated delays in Germany or in the EU. These reasons may be less relevant for people working in other contexts, but this is not a limitation in qualitative research.

## 5. Conclusions

A wide range of 66 reasons could be identified empirically (Table 3). This spectrum was used to identify and further analyze ethically relevant aspects. It can be used by researchers and in training to reflect on decision-making. The range and ethical discussion can provide illustrative material for this, although it is important to note that this paper is not a training manual.

Ethical frameworks were useful to systematically address the ethical dimensions, mainly those of animal research ethics. Regarding the ethical analysis, the reasons were then reconstructed into relevant descriptive and evaluative premises according to a *Value Judgement Model* [23]. All considerations in ethical decision-making have to be weighed against each other and accordingly balanced in a conclusion. Thereby, considerations based on generally accepted values, such as the principle of not harming or killing, should be assumed to be of fundamental importance (“ethically superior”). Such values should be given more weight than, for example, the desire to gain media attention. This demonstrates that the quality of the reasons and not the quantity is important in the weighing process. The weight we give to a particular consideration in the decision-making process also depends on the certainty accompanying it. However, the central moral problem consists of the conflict between *animal welfare* and (expected) *social benefit* when using animals in biomedical research (e.g., [28]). There are serious ethical arguments on both sides in this classic ethical dilemma. This means that the rejection of alternatives can also have an ethical justification (e.g., “protecting humans from ineffective or harmful drugs is a higher value than not using animal models,” PA2.4).

Researchers should be supported in the decision-making process because there are various aspects to be considered in balancing (all aspects relevant and more or less legitimate reasons could be involved). First of all, this means raising the awareness of decision-making situations. Even if it sometimes seems that everything is predetermined, decisions are always being made about a particular model in a specific project, about the strategic engagement with a new model/technology, or even about the long-term orientation of a research team (i.e., whether or not the team intensifies the usage or development of alternative methods). Appropriate measures to support conscious decision-making range from documentation sheets for ethical decision-making to training courses and institutionalized ethics counselling; however, they must not lead to more paperwork being required and make practical work more difficult.

There are also many dependencies in science (e.g., young scientists towards not only senior researchers, but also funders). This results in power structures that should not be a burden for ethical decision-making or a slowdown of further development.

Currently and in the mid-term, however, no research model or, more specifically, disease model, represents the human life form in the same way as the real human organism. As long as animal experimentation is not prohibited in principle (and decision-making is, therefore, unnecessary), this means that researchers rely on various value judgements that are at stake and should refer to broadly accepted criteria for choosing a research model.

## Figures and Tables

**Table 1 animals-14-00651-t001:** Three value judgments and their premises according to the Value Judgment Model.

Value Judgment		Evaluative Premises	Descriptive Premises
“Alternative method A has less serious consequences (especially for animals) if the experiment fails than animal experiment B” *	Premise	Serious consequences (especially for animals) when experiments fail are bad (*should be avoided*) °	If experiments fail, alternative method A has consequences Z (especially for animals) °
	Backing Premise	Animal welfare (*value*), efficiency (*value*) °	The animal experiments in this kind of research imply problematic consequences °
Alternative method A is not cruel (compared to the animal experiment B) °	Premise	“The animal experiments (in this kind of research) were cruel” *	Alternative method A does not imply that Z has to be done to animals (e.g., inducing strokes in rats) °
	Backing Premise	Animal welfare (*value*) °	For investigating this topic, it is necessary to do Z to animals (epistemic processes) °
Alternative method A better complies with the demands of the society, and/or better avoids societal criticism °	Premise	To comply with what the society demands more, and/or to avoid societal criticism, is favourable °	Alternative method A is accepted better by society (fulfils its demands), and/or is less/not criticized °
	Backing Premise	Democracy/participation (*value*), alignment with society (*interest*), avoidance of criticism (*emotion*) °	“Society demands more alternatives, and animal experiments are criticized even more as before” *

* Quote from interviews (abstracted for anonymization purposes) ° Reconstructed by the authors.

**Table 2 animals-14-00651-t002:** Characteristics of the experts interviewed.

Characteristics	Sample (*n* = 13)
Gender (w/m)	31% women
Age (years)	8% under 30 y.|46% between 40 and 50 y.|46% over 50 y.
Level of expertise (junior/senior)	62% senior researcher
Kind of experience (model)	23% only animal|15% only non-animal models|62% both
Affiliation (academic/industry)	100% academic
Country (work place)	100% Germany
Length of interview (min)	20 to 46, mean 32

**Table 3 animals-14-00651-t003:** The 66 reasons why peoples choose an alternative or animal model.

Area of Reasons	I Choose the Alternative Because …	I Choose the Animal Model Because …
Personal Attitudes (PA)	PA1.1: “I am curious to try new things” (E)	PA2.1: “It can be justified and is ethically acceptable” (E)
	PA1.2: “it draws (media) attention” (D) PA1.3: “experiments with animals are stressing me emotionally/psychologically” (E) PA1.4: “animal experiments (in this field) were cruel” (E) PA1.5: “I want to contribute to change the current practice in research and development” (E)	PA2.2: “I will/must comply with the ‘state of the art’ (animal experiments)” (E) PA2.3: “I find it important to carry out experiments again myself and, thus, confirm known results” (E) PA2.4: “protecting a human being from ineffective or harmful drugs is a higher value than refraining from animal experiments” (E) PA2.5: “I lack confidence in the potential of alternatives” (E) PA2.6: “I feel that this is my responsibility in the education of veterinarians (E)
Work Environment (WE)	Reasons Concerning Work Organization	
	WE1.1: “alternatives are cheaper” (E) WE1.2: “alternatives are associated with smaller amounts of lengthy bureaucracy (e.g., no approval procedure)” (D) WE1.3: “alternatives can be realized more quickly” (D) WE1.4: “there is suitable infrastructure on site” (D) WE1.5: “I am not authorized and/or qualified to conduct animal experiments” (E) WE1.6: “animal keeping is quite costly” (E) WE1.7: “we never would have been able to do that quantity of tests on animals” (D) WE1.8: “I get more freedom (in choosing experimental design, methods) than in animal experiments” (E)	WE2.1: “there are too many research ethics and legal requirements for research on humans or human tissue” (E) WE2.2: “the development of alternatives is longsome and inadequately funded (while animal experiments are established)” (E) WE2.3: “alternatives are often more expensive than an animal experiment” (E)
	Research Climate within Institution	
	WE1.9: “superiors are demanding or are explicitly supporting the use of alternative methods” (D)	WE2.4: “my supervisor has decided so” (D)
	Expert Opinion and Research Funding	
	WE1.10: “funding lines exist exclusively for them” (D) WE1.11: “it can be used to answer the now wider-ranging and more complex research questions” (D) WE1.12: “reviewers respond in a constructive manner” (D) WE1.13: “I was contacted by agencies/research funders to use (test, develop) alternatives” (D)	WE2.5: “reviewers tend to come from a generation in which animal experiments are recognized above all” (D)
	Peer Group/Scientific Community	
	WE1.14: “I have already internalized the 3R and am, therefore, more receptive to associated innovations” (E) WE1.15: “alternatives have, in the meantime, been accepted by the community” (D)	WE2.6: “science has 50 to 60 years of experience with certain animal models” (D)
	Education and Teaching	
	WE1.16: “I would like to reduce the number of animals required for educational purposes” (E)	WE2.7: “there is some content that simply could not be communicated without the direct use of animals” (D) WE2.8: “animal experiments are legally part of the training of veterinarians; it is written in the license to practice and, therefore, mandatory” (D)
	Society	
	WE1.17: “society is demanding more alternatives and animal experiments are respectively more criticized than in the past” (D)	---
	Technical Development	
	WE1.18: “new milestone technology (e.g., CRISPR/Cas, IPSC) is enabling me to work in a more targeted manner on a genetic level than working with animal models (e.g., mice)” (E) WE1.19: “I have learned about their many uses through previous experiments with alternatives” (D)	WE2.9: “alternative methods are not yet fully developed in my field” (D) WE2.10: “I can’t (yet) connect the correspondences of the organs/functions in alternatives (so I can’t test complex interactions)” (D)
Animal (A)	Animal Welfare/Dignity	
	A1.1: “I 1: “I want to avoid animal suffering” (E) A1.2: “it enables various pre-experiments that can avoid animal experiments which otherwise would have to occur” (D) A1.3: “The consequences of a failed experiment are much less critical than in animal experiments (especially regarding the animal)” (E)	A2.1: “they are necessary and before someone does it who doesn’t care about animals, I prefer to do it myself” (E)
Science (S)	Research Questions and Approaches	
	S1.1: “a correct replication of the disease or (e.g., cellular) processes cannot be obtained through the animal model (e.g., mouse)” (D) S1.2: “I can have a greater degree of control (manipulability) over the experiment” (D) S1.3: “I can preselect (narrow down) substances this way” (D) S1.4: “I can better describe and/or explain the basic effects/mechanisms here” (D) S1.5: “I find the transferability of animal models to humans inadequate” (E) S1.6: “I can get results without hypotheses” (E) S1.7: “I only want to observe a certain step/function (e.g., effector function of T cells against tumors) and not the whole process (as would occur in an animal)” (E)	S2.1: “experiments on living animals rather allow one to discover completely new (unexpected) things” (E) S2.2: “complex questions/interactions can only be investigated in the whole organism” (D) S2.3: “I perform research in animals for animals of their species (veterinary medicine), which is very difficult to replace with alternative methods” (E) S2.4: “to gain access to and expertise in a disease from it” (D)
	Results	
	S1.8: “the standardizability of the experiment is higher (more valid and reproducible results)” (E) S1.9: “it allows me to avoid variability between individual animals” (D) S1.10: “conditions of animal keeping in the laboratory can distort the results of animal experiments, and some of these effects are unknown” (D)	S2.5: “alternatives currently produce too many false-positive or false-negative results” (E)
	Translation/Usability	
	S1.11: “it is now increasingly possible to obtain approval for new experimental therapeutic approaches without prior animal testing” (D)	S2.6: “the industry often demands results from animal models, and otherwise commercialization of our results is hardly possible” (E) S2.7: “the results of alternatives (alone) are an insufficient basis for clinical studies with humans (translation)” (E) S2.8: “it is a test method recognized by the OECD in safety assessment” (TOX) (D) S2.9: “there are also legal requirements that prescribe some animal experiments” (D)
	Publications	
	S1.12: “I get publishable results faster when I am using an alternative” (E)	S2.10: “because with alternatives you have a hard time with some important journals” (E)

(E) = reason refers to an evaluative premise/backing or to the value judgment itself; (D) = refers to descriptive premise/backing.

## Data Availability

The audio and transcripts of the interviews have not been authorized for publication by the interviewees. For reasons of anonymity, they cannot be published either.

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
