# Peer review of "Why Do I Choose an Animal Model or an Alternative Method in Basic and Preclinical Biomedical Research? A Spectrum of Ethically Relevant Reasons and Their Evaluation"

_animals, 2024, doi:10.3390/ani14040651_

Round 1
Reviewer 1 Report
Comments and Suggestions for Authors
Kahrass et al. describe an interesting and relevant study of the reasoning behind methods selection in human preclinical science. The manuscript is generally well-written and a pleasant read, and should be acceptable for publication in “Animals” after minor revisions. These revisions should address my two concerns: some parts need nuancing (detailed below), and the qualitative methods need to be explained at a level that is fully comprehensible for those working in more quantitatively oriented fields, which comprises the main readership of “Animals”.
Specific comments:
L10-11: The first sentence is no great start; the focus is on preclinical-human (not preclinical – veterinary), which should be clear from the start, and “either animals or alternative methods” is hardly ever a properly dichotomous choice (if an alternative is available, it should be used, but research questions can be and regularly are changed to end up with the preferred method). It would be helpful to define “alternative methods” before introducing the choice; does the term “alternative method” in this paper comprise e.g. in silico and desk research?
L24-25: as above.
L52: as above.
L68-69: May need rewriting depending on the definition of “alternative method”
L94-95: “basic” and “translational” research need to be defined and the difference in structure needs to be explained.
L178: “selection criteria” suggests hard yes/ no criteria while the surrounding texts suggests that these factors were used to create a varied sample. Please clarify.
L290-317: Will these results be analysed further (e.g. in a separate paper)? This would definitely be interesting.
L199-213: please add additional information for those not familiar with qualitative research.
L288-289 (Table 1): The later interpretation would benefit from some additional description of the sample, and focus on the confounding caused by seniority. The field is definitely more female than the sample with only 31% women. Adding a paragraph about the potential gender effect to the discussion is not essential, but it would be valuable.
L383-386: It would be valuable to refer to the literature on “compassion fatigue” here.
L416-418: It might be valuable to refer to the literature on (mutual) dependence in scientific hierarchies.
L441: certainly not “always” (and also please be aware that misconceptions about differences in “transferability” between fields are common).
L506: should “more” in this phrase not be “less”?
L511-512: this is highly unlikely; animal models are generally a lot more expensive than any conceivable alternative.
L536-537: you should not say “the largest quantitative share” without defining the total. In this case I’d prefer different phrasing, as it is more about the number of different reasons within a category, while there is no proper quantification of the categories as a whole.
L602-603: Either delete the sentence about “retroactive consideration” or add nuancing; your sample is mainly old and from a time that alternative methods were not available. Retroactive consideration thus is hardly relevant.
L532 and 618: There is a surprising lack of actual evidence for “social benefit” of animal experiments. I’d suggest to delete the term from the manuscript or somehow rephrase (“supposed benefits”).
Comments on the Quality of English LanguageL337: Delete the “s” after “people”
Author Response
Reviewer 1:
L10-11: The first sentence is no great start; the focus is on preclinical-human (not preclinical – veterinary), which should be clear from the start, and “either animals or alternative methods” is hardly ever a properly dichotomous choice (if an alternative is available, it should be used, but research questions can be and regularly are changed to end up with the preferred method). It would be helpful to define “alternative methods” before introducing the choice; does the term “alternative method” in this paper comprise e.g. in silico and desk research?
Authors: Thanks for the remark. We have rewritten the beginning of the simple summary: “Some basic and preclinical biomedical research models require the use of animals. It is not always clear which model is best suited to a particular project – animals or models based on e.g. in vitro or in silico methods?”.
We also added again “biomedical” to emphasize that not veterinary research is meant; later in the main text, we included the following remark: “biomedical research (i.e. research that aims to benefit, in the end, human health)”. Thus, for us, the term “biomedical” is used to describe the intention to serve humans; we could not find definitions that (at least expressively) subsumes veterinary medicine under the terms “biomedical research” or “biomedical medicine”.
We decided to define “alternative methods” more in detail later, as the place in the simple summary and abstract is quite limited (see below).
L24-25: as above.
Authors: We changed it to “Therefore, this paper aims, (1) to identify a spectrum of reasons for choosing between animal and non-animal disease research models (e.g. based on in vitro or in silico models) […]”. (For these changes, however, we had to shorten the abstract somewhat in other places).
L52: as above.
Authors: We changed it accordingly also in the main text to “[…] reduced necessity due to increasing alternative methods such as in vitro or in silico methods […]”.
L68-69: May need rewriting depending on the definition of “alternative method”
Authors: We changed and added the following:
“The project is based on the assumption that the researcher has three basic options: use (also) an animal model, use solely alternatives or forego research. In normative terms, the decisions made must be based on good (= sufficiently justifiable) reasons – be they ethical, epistemic/scientific or, as the case may be, merely pragmatic. The many small decisions about how to approach a particular research question, or even just one aspect of it and the reasons for it, will be the focus of the following article. In this context, ‘making choices’ refers to the many small decisions about how to approach a particular research question, or even just one aspect of it. Countless such choices can be made in research projects, in working groups, and even more so in a researcher’s career. As researchers usually pursue many research questions at the same time (have different experiments running), they often work with animal and non-animal models in parallel [REF].”
We also added a more clear description of alternative methods:
“Alternatives roughly refer to all approaches that replace animals or substantially reduce their use in the research context. Some call them ‘new models’ [REF], meaning, for example, employing in vitro 2D or 3D cell cultures, in silico methods and new milestone technology (e.g. CRISPR/Cas, IPSC). To a certain extent, this can also include desk research methods such as systematic reviews if their results lead to a reduction in the number of animal experiments in the future [REF].”
L94-95: “basic” and “translational” research need to be defined and the difference in structure needs to be explained.
Authors: Thanks. We have decided to use preclinical consistently and have deleted translational, as it proved somewhat difficult to categorize the interviewees here anyway. We are aware that there are significant differences between basic and preclinical and translational. However, at this particular point we are only interested in the demarcation from toxicology.
L178: “selection criteria” suggests hard yes/ no criteria while the surrounding texts suggests that these factors were used to create a varied sample. Please clarify.
Authors: Many thanks for the comment. We realized that we were a bit unclear and have rewritten the text. The first three criteria should now be more clearly recognized, i.e. “hard” inclusion criteria as you described (now: “The “hard” inclusion criteria were […]”). Furthermore, it now says: “Further categories applied to define the sample were different career levels (PhD to senior researcher), affiliation and gender.”
L290-317: Will these results be analysed further (e.g. in a separate paper)? This would definitely be interesting.
Authors: This section is about different understandings of alternatives. We highlight differences in understanding and discuss what is possible based on n=13 qualitative interviews. Our results are fully reported at this point and we see no opportunity to analyze them further.
L199-213: please add additional information for those not familiar with qualitative research.
Authors: The passage was rewritten. It now says:
“At the beginning, the interview transcripts are carefully read line by line, marking passages entailing value judgments and the associated argumentation, using the MAXQDA 2018 software. The content was then characterized using existing concept- (deductive) and text-driven/generated (inductive) codes or ‘headings’ (e.g. time commitment, trust in alternatives, or career with a subheading on: pressure from superiors but also prestige). The concept-driven elements were based on a certain understanding of value judgements and their structure developed within the project, the so-called Value Judgment Model (see below). This is necessary because in the empirical analysis step, value judgments are theoretical constructs, i.e. they cannot be directly empirically ascertained. In order for value judgements and their components and justifications (e.g. in the form of descriptive assumptions, but also values, interests or emotions etc.) to become recognizable (“visible”) in a text, an “operationalization”, here in the form of a conceptual model, must therefore be used.
The text-driven code generation was successively reconciled with the more theoretical-driven coding. Once the content of all interviews has been coded, the codes were compared, and similar codes were merged into paraphrases that reflect the core of a stated reason for a decision. These paraphrased reasons are thus a condensed reproduction of the statements from the interviews, which is accompanied by a certain degree of abstraction. This allows concise reasons to be extracted from the interviews, based on multiple passages from multiple people, making them less subjective.”
L288-289 (Table 1): The later interpretation would benefit from some additional description of the sample, and focus on the confounding caused by seniority. The field is definitely more female than the sample with only 31% women. Adding a paragraph about the potential gender effect to the discussion is not essential, but it would be valuable.
Authors: Thank you for the suggestion! We added 7 characteristics about the sample in Table 2 (old table 1) and address the dominance of senior male researcher in the limitations section. There, it now says:
“The theoretical saturation in this project was reached; nevertheless, it is important to note that we might have missed reasons that could, for example, by using of different ethical theories or approaches. Regarding the latter, we have not analysed or discussed (therio)epistemological issues, and have decidedly not taken a perspective from the philosophy of science, as one of our colleagues in R2N did [REF]. Our focus was on the ethical value judgments and associated reasons. However, we have implicitly addressed epistemological reasons insofar as they are presupposed in certain ethical norms. For example, in norms that demand scientific validity; but basically already in the Principle of No Alternatives, which presupposes the answer to the epistemological questions of what knowledge can be generated by animal experiments or alternative methods and how “good” this knowledge is. Furthermore, other reasons might have been found in a sample with more female respondents. In our sample, senior male researchers in Germany where the most dominant group. The spectrum presented can and should therefore be complemented by further qualitative research. However, it is important to notice that qualitative research strives for diversity and range of topics and not representativeness in a statistical sense. On a personal level, not all 66 reasons are likely to be equally important, and some will not resonate at all. Our qualitative approach does not allow us to say anything about the frequency of the reasons given. It is possible that some reasons are primarily due to particularities such as the bureaucratic effort and associated delays in Germany or in the EU. These reasons may be less relevant for people working in other contexts, but this is not a limitation in qualitative research.”
L383-386: It would be valuable to refer to the literature on “compassion fatigue” here.
Authors: Many thanks for the tip. We agree that this is an obvious topic. However, we cannot go into it in detail, nor can we cite much literature, as we have been asked by the editors to be sparing with the literature. We have therefore decided to make a brief reference to the topic and to refer to some exemplary literature which, in our opinion, deals with the topic broadly enough:
“In addition, if someone argues that conducting animal experiments stresses them emotionally – a phenomenon that is now well known and is treated under terms such as “compassion fatigue”, “mental stress” or “moral stress” [e.g. REF] –, this probably (implicitly) refers to a general norm of non-maleficence, such as “do not harm others” or “thou shalt not kill”.
L416-418: It might be valuable to refer to the literature on (mutual) dependence in scientific hierarchies.
Authors: Thank you also for this hint. As this would open up a whole new can of worms from the sociology of science (with the corresponding explanations required), which is also not as directly illustrative for animal research as compassion fatigue, we have decided to refrain from including it.
L441: certainly not “always” (and also please be aware that misconceptions about differences in “transferability” between fields are common).
Authors: We agree and changed the sentence: “Reasons belonging to this part of the spectrum are based on statements about specific fields of research.“
L506: should “more” in this phrase not be “less”?
Authors: We recognize that the wording might have been confusing, so we have reworded the sentences as follows: “However, accepting such a premise obviously opens up countless exceptions to the Principle of No Alternative Method. Thus, if alternatives would be available or could conceivably be developed, insufficient funding is not an ethically convincing argument for the use of animal models.” (Please note that due to other reviewer comments, the entire text passage containing this sentence has been revised).
L511-512: this is highly unlikely; animal models are generally a lot more expensive than any conceivable alternative.
Authors: We added a sentence to specify our statement. It now says: “As the resources are allocated in different funding lines, it may be the case that no funding is available for the development of infrastructure and the qualification of staff for research with alternative methods, but funding is available for the acquisition of a certain number of mice.“ (Again, please note that due to other reviewer comments, the entire text passage containing this sentence has been revised).
L536-537: you should not say “the largest quantitative share” without defining the total. In this case I’d prefer different phrasing, as it is more about the number of different reasons within a category, while there is no proper quantification of the categories as a whole.
Authors: Thanks. We changed the wording: “A considerable amount of reasons can be found in the categories belonging to work environment (> 37 %) and the scientific context (> 29 %)”. As the statement is based on percentages, we consider this to be appropriate. The total number are the 66 reasons mentioned one sentence earlier.
L602-603: Either delete the sentence about “retroactive consideration” or add nuancing; your sample is mainly old and from a time that alternative methods were not available. Retroactive consideration thus is hardly relevant.
Authors: The text about limitations was generally revised and, in the course of this, we deleted the criticized text passage.
L532 and 618: There is a surprising lack of actual evidence for “social benefit” of animal experiments. I’d suggest to delete the term from the manuscript or somehow rephrase (“supposed benefits”).
Authors: Thanks for the comment, but we disagree regarding the rephrasing. “Social benefit" is a fixed term in the principlism approach used, and thus refers to an ethical principle and underlying values (a similar term with similar meaning, “social value”, is used in a principlism approach for clinical research ethics). The question of whether and to what extent animal experiments realize this principle or value is another matter. In fact, this ethical principle is needed precisely in order to be able to say critically that a specific expected or actual outcome of an animal experiment cannot justify the harms inflicted. Social benefit is always to be understood in the sense of “expected” or “hoped-for” benefits in the context of, for example, a prospective harm-benefit analysis.
To clarify this point, however, we have added “expected”, i.e. “(expected) social benefit”.
L337: Delete the “s” after “people”
Authors: Thanks, corrected.
Reviewer 2 Report
Comments and Suggestions for Authors
Review Animals-2812139
This paper reports an interview study with biomedical researchers about their reasons to choose different types of models for research. The study seems to be well done, although the small sample size (13 researchers) is a limitation. The topic is timely, and very little research has addressed the important question of how researchers make these choices, an issue that is critical for the implementation of the 3Rs and in particular the transition towards non-animal methods. I appreciate the opportunity to review this work and here indicate issues that need addressing in a revision:
Major issues
1. The strong focus on the ethical dimension of the question under study means that the authors lose sight of the equally important epistemological dimension. On the one hand, that means that the paper doesn’t really do justice to the scholarly literature on non-animal models in research. In particular the section on lines 58-72 should present the state-of-the-art knowledge with reference to the work that has been done on the topic. In the end of this review, I give a (not exhaustive) list of relevant and fairly recent papers. On the other hand, the relevance of the ethical dimension is sometimes overstated. In particular, to say that “ethics as the systematic exploration of values, norms (…) provides the relevant theoretical and methodological background” (lines 112-115) is an overstatement. Ethics is certainly one relevant approach to this, but epistemology would be equally relevant and complementary. Several of the papers listed below are taking an epistemological approach to the question. This needs to be discussed.
2. Please provide more information about the interviewees: country of work, field of research, type of animals, in which type of organization do they work.
3. There are two sections in the narrative presentation of the results which are not well aligned with the tabulated presentation of reasons. The narrative presentation of the sources of funding seems to be about using lack of funding as a justification for continuing to use animals in experiments, but this is not what I read the two cited reasons to be about. WE1.10 and WE 1.13 seem to be about researchers working with the development of alternatives because there is funding for that. The narrative presentation of the animal welfare category is very abstract and difficult to read; linking it to results in Table 2 (as is done for the other categories) would help to make it less abstract.
4. Related to point 1 above: the first paragraph in the Discussion says that the reasons reported by researchers are of a type that receive less attention in the ethical debate. This may be true for the academic debate of animal research ethics, but there is a political and societal discussion of animal experimentation where these questions and especially these of the scientific context are prominent. This part of the debate should not be overlooked in the paper.
Minor issues
Lines 10-11 That researchers can use either animals or alternative methods is a huge oversimplification and it’s unfortunate to state this in the simple summary as it reinforces a common misunderstanding. Most biomedical researchers use both animals and non-animal methods, depending on what research question they are addressing.
Line 15 I understand that with “ethically analysed” the authors want to say “analysed using an ethics approach”, but this is not how the word “ethically” is usually used.
Line 17 What does “can only be influenced to a limited extent” mean? Influenced by whom?
Lines 29-30 Need to say that these are the reasons researchers refer to as underlying their choices.
Lines 60-61 I would insert “an externally imposed” before “regulatory and often related ethical obligation” to justify the statement that these obligations are increasing. If it’s ethically wrong to harm animals for human benefit, this would have been as ethically wrong in 1994 as it is in 2024.
Line 87 A “disease model” is a model of a disease, and is a term that is relevant in biomedical research but not in toxicology which is the context here. Suggest to replace with “test method”.
Lines 92-93 Suggest to add “and required” to become “or the animal test is permitted and required”
Lines 95-96 “and have a wider variation, for example, whether a chemical substance causes measurable skin irritation” – something is wrong with this sentence which is about biomedical research but uses a toxicology example.
Line 194 “The insights we gained (…) are based primarily on individual experiences and personal attitudes” – this refers to the individual experiences and personal attitudes of the interviewees, right? This need to be said.
Lines 234-236 In the description of the value judgement model, it’s said that value judgements are the conclusion of an argument that consists of at least one descriptive premise and at least one evaluative premise. Does this mean that the excerpts of the interview needed to include at least one descriptive premise and at least one evaluative premise? Please clarify.
Line 260 “The animal research ethics approach” is used here as if there is a universally recognised concept for how to approach animal research ethics, which is based on a set of principles. I think the principles listed here are a good description of the basic aspects of mainstream animal research ethics, but they are in no way as universally recognised in animal research ethics as the ones by Beuchamp and Childress are in medical ethics.
Table 2: WE 1.10-WE1.13 are presented twice.
Lines 412-414 Unfortunate wording. It’s not bureaucracy (either in the meaning of a governance system where public servants make many decisions or in the meaning that the interviewee refers to, as a complicated system) that is a best practice to ensure validity and integrity. Rather say something like “The review system for animal experiments, which may be perceived as burdensome bureaucracy, is not an end in itself”.
Line 447 suggest to revise to “seems to be more obvious in some fields and for some research questions”.
Lines 480-482 cite Garner et al (2017; see below) who write precisely about the kind and scale of uncertainties are assumed when working with animal models
Davies G, Greenhough B, Hobson-West P, Kirk RGW. Science, culture, and care in laboratory animal research: interdisciplinary perspectives on the history and future of the 3Rs. Sci Technol Hum Values. (2018) 43:603–21. doi: 10.1177/0162243918757034
Grimm H, Biller-Andorno N, Buch T, Dahlhoff M, Davies G, Cederroth CR, Maissen O, Lukas W, Passini E, Törnqvist E, Olsson IAS and Sandström J (2023) Advancing the 3Rs: innovation, implementation, ethics and society. Front. Vet. Sci. 10:1185706.doi: 10.3389/fvets.2023.1185706
Garner, J., Gaskill, B., Weber, E. et al. Introducing Therioepistemology: the study of how knowledge is gained from animal research. Lab Anim 46, 103–113 (2017). https://doi.org/10.1038/laban.1224
Herrmann K, Pistollato F, Stephens ML. Beyond the 3Rs: expanding the use of human-relevant replacement methods in biomedical research. ALTEX. (2019) 36:343–52. doi: 10.14573/altex.1907031.
Lowe et al 2018 Training to Translate: Understanding and Informing Translational Animal Research in Pre-Clinical Pharmacology https://www.animalresearchnexus.org/publications/training-translate-understanding-and-informing-translational-animal-research-pre
Comments on the Quality of English Language
The language is acceptable but getting a professional review of the English would improve readability. The writing is quite clunky in places, starting in the very beginning of the introduction, where the second sentence is a striking example: "It is also considered to be a controversial social and political issue due to housing conditions, the (sometimes perceived) reduced necessity due to increasing non-animal alternatives and generally the fact that animals are subjected to stress, pain and even death."
Author Response
- The strong focus on the ethical dimension of the question under study means that the authors lose sight of the equally important epistemological dimension. On the one hand, that means that the paper doesn’t really do justice to the scholarly literature on non-animal models in research. In particular the section on lines 58-72 should present the state-of-the-art knowledge with reference to the work that has been done on the topic. In the end of this review, I give a (not exhaustive) list of relevant and fairly recent papers. On the other hand, the relevance of the ethical dimension is sometimes overstated. In particular, to say that “ethics as the systematic exploration of values, norms (…) provides the relevant theoretical and methodological background” (lines 112-115) is an overstatement. Ethics is certainly one relevant approach to this, but epistemology would be equally relevant and complementary. Several of the papers listed below are taking an epistemological approach to the question. This needs to be discussed.
Authors: Thank you very much for the, of course, justified objection, and for the literature. However, we are unable and, to a certain extent, also unwilling to fully reflect the epistemological perspective in addition to the ethical perspective. There are project-related, institutional, and content-related reasons for this.
On the one hand, it was already a conscious decision at the application stage to focus one ELSI-project in R2N (E1, ours) on ethics. The second ELSI-project (E2, colleagues), on the other hand, focused more strongly on epistemological aspects and a philosophy of science perspective from the outset. It was therefore clear from the start that we would not be working on this topic. Also, the funding for the project has long since expired. We no longer have the resources available to comprehensively work on the epistemological aspects – and definitely not so within a short revision period of a paper!
Nevertheless, we tried to make now clearer that the focus is on ethics. We changed the title slightly to “A spectrum of ethically relevant reasons and their evaluation” and changed a bit in the introduction section (Aims and Research Context) to reflect this project-related and institutional circumstances: “Selected results from one of the ELSI-projects are presented here; related preliminary theoretical work has already been published in [REF]. Work from the second ELSI-project that focused more on a perspective from philosophy of science has also been published [REF].” We have also made minimal changes to the Simple Summary and the Abstract to make it a little clearer that it is about ethical decision-making.
More importantly, however, it is also possible to discuss whether the reviewer’s assessment is correct. Of course, we respect her/his view. We, however, like to disagree. We do this primarily because, in our view, many of the epistemological questions ultimately follow from an ethical norm of scientific validity or a scientific ethos on the one hand (cf. also Merton’s scientific ethos and the references there to quasi-moral imperatives in scientific norms), and from norms that aim to prevent or reduce animal suffering on the other.
It goes without saying that the epistemological questions are not answered by referring to ethical norms. That needs its own discussion. However, we would clearly defend the assumption that these epistemological questions are only asked when choosing between animal experiments and alternatives because they are, in the end, ethically relevant. In other words, from our perspective, epistemological questions are “always already contained” in at least those ethical norms that aim for scientific validity (however, also the Principle of No Alternative in fact entails epistemological questions, as if one invokes this principle in a concrete case, it also means that there has to be an argument that there are no viable alternatives, which in turn will refer to epistemological arguments). Therefore, the discussion of epistemological questions has an ethical significance – which is why, in our opinion, ethics does indeed provide a “relevant theoretical and methodological background” when it comes to such decision-making.
But of course, as is usual in the humanities and social sciences, one can also put on other “glasses” to look at decision-making processes. We therefore did not want to claim that there are no other relevant backgrounds or approaches possible, and have weakened the wording accordingly (see below). Since we do not deny that the epistemological questions require an independent discussion, we have endeavoured to address the topic in three places in the text (but, as said, not in detail):
1) Introduction (section Ethical and regulatory background for using animals in basic and preclinical research):
“In addition, there are well-known objections about the extent to which animal experiments are even transferable to humans and/or how much they can advance scientific knowledge in the biomedical field [REF]. Whether and to what extent robust knowledge can be generated from animal experiments, and to what extent this knowledge can also be significant for the development of diagnostics or therapies in humans, are epistemological questions, or questions for philosophy of science [e.g. REF]. The term Therioepistemology was coined for this a few years ago [REF]. Such questions are often inevitably asked in comparison with the question of the extent to which this knowledge can be achieved equally or even better through alternatives, especially in the case of human-relevant alternatives, e.g. patient-derived cells [REF]. Although such epistemological questions can also be addressed independently of ethical considerations, it is also argued that animal experiments which do not have sufficient scientific value fail to be ethically acceptable even if all other ethical requirements are fulfilled; Strech and Dirnagl, for example, propose three principles to safeguard and enhance the scientific value of animal research: robustness, registration and reporting [REF]. In the end, thus, questions related to robustness and value of knowledge gained from animal research or alternative methods almost always take place in an ethically sensitive discourse in which ethical norms such as the obligation to generate "social benefit" with animal experiments – which is only possible through scientifically valid research – and norms aiming at the avoidance, or at least reduction, of animal suffering can be identified in the background.”
2) Introduction (Section Decision-making and the role of ethics):
“Thus, it is often not clear in basic and preclinical research whether the use of a particular alternative method is really a valid alternative, i.e., if it leads to comparable results to the animal experiment, allows the testing of the same hypothesis or even maintaining the original research question; the epistemological questions are therefore not always answered. […] In this context, ethics, as the systematic exploration of values, norms and principles, and as the critical examination of arguments involving (moral) value judgments, provides the a relevant theoretical and methodological background for analyzing and evaluating decision-making – which does not preclude other relevant approaches, such as those from (therio)epistemology or cognitive psychology.”
3) Discussion, part where we talk about limitations of the study:
“The theoretical saturation in this project was reached; nevertheless, it is important to note that we might have missed reasons that could, for example, by using of different ethical theories or approaches. Regarding the latter, we have not analysed or discussed (therio)epistemological issues, and have decidedly not taken a perspective from the philosophy of science, as one of our colleagues in R2N did [REF]. Our focus was on the ethical value judgments and associated reasons. However, we have implicitly addressed epistemological reasons insofar as they are presupposed in certain ethical norms. For example, in norms that demand scientific validity; but basically already in the Principle of No Alternatives, which presupposes the answer to the epistemological questions of what knowledge can be generated by animal experiments or alternative methods and how “good” this knowledge is.”
Lines 10-11 That researchers can use either animals or alternative methods is a huge oversimplification and it’s unfortunate to state this in the simple summary as it reinforces a common misunderstanding. Most biomedical researchers use both animals and non-animal methods, depending on what research question they are addressing.
Authors: Rewritten. It now says: “Some basic and preclinical biomedical research models require the use of animals. It is not always clear which model is best suited to a project – animals or models based on e.g. in vitro or in silico methods?” In the introduction, we further added: “In this context, ‘making choices’ refers to the many small decisions about how to approach a particular research question, or even just one aspect of it. Countless such choices can be made in research projects, in working groups, and even more so in a researcher’s career. As researchers usually pursue many research questions at the same time – have different experiments running –, they often work with animal and non-animal models in parallel [10].”
Line 15 I understand that with “ethically analysed” the authors want to say “analysed using an ethics approach”, but this is not how the word “ethically” is usually used.
Authors: As ethicists, we would like to take the liberty of slightly disagreeing here regarding the meaning and use of “ethically analyzed”. It is not uncommon in ethics to speak of “ethically analyzed” to describe an analysis from an ethical point of view – just as, for example, “sociologically analyzed” or “psychologically analyzed” means that something is analyzed from a sociological or psychological point of view. “Ethically” here is a reference to the discipline of ethics, not an adjective with the meaning “in a moral way” or similar.
However, we acknowledge that people who are not used to the language of ethics (as a discipline) may stumble over this. We have therefore reworded the passage as follows: “The responses were qualitatively assessed and subjected to an ethical analysis”.
Line 17 What does “can only be influenced to a limited extent” mean? Influenced by whom?
Authors: Thanks for the remark. We added “by individuals“.
Lines 29-30 Need to say that these are the reasons researchers refer to as underlying their choices.
Authors: We changed it to “This paper presents 66 reasons underlying the choice of researchers using animal (41 %) or alternative models (59 %).”.
Lines 60-61 I would insert “an externally imposed” before “regulatory and often related ethical obligation” to justify the statement that these obligations are increasing. If it’s ethically wrong to harm animals for human benefit, this would have been as ethically wrong in 1994 as it is in 2024.
Authors: Here one can take different positions with regard to normative-ethical or meta-ethical positions (regarding universalism/absolutism and relativism), but we agree with the addition of “an externally imposed”, as suggested.
Line 87 A “disease model” is a model of a disease, and is a term that is relevant in biomedical research but not in toxicology which is the context here. Suggest to replace with “test method”.
Authors: This is a good and, of course, correct observation, thanks a lot. We changed it accordingly.
Lines 92-93 Suggest to add “and required” to become “or the animal test is permitted and required”
Authors: Thanks; we changed it accordingly.
Lines 95-96 “and have a wider variation, for example, whether a chemical substance causes measurable skin irritation” – something is wrong with this sentence which is about biomedical research but uses a toxicology example.
Authors: That is right. The comparison with toxicology, which of course belongs here so that the sentence makes sense, has been missing. We have rewritten the sentence. It now says: “On the other hand, the research questions and objectives in basic and preclinical research are different and more variable than, for example, testing whether a chemical substance causes measurable skin irritation, as part of a toxicology testing.”
Line 194 “The insights we gained (…) are based primarily on individual experiences and personal attitudes” – this refers to the individual experiences and personal attitudes of the interviewees, right? This need to be said.
Authors: Correct; we changed it accordingly.
Lines 234-236 In the description of the value judgement model, it’s said that value judgements are the conclusion of an argument that consists of at least one descriptive premise and at least one evaluative premise. Does this mean that the excerpts of the interview needed to include at least one descriptive premise and at least one evaluative premise? Please clarify.
Authors: Ideally, this would have been always the case; in Mertz et al 2023 [new ref no. 25], we discuss the problem of exciting in an interview the premises relevant for such a model, and that this could be improved in a future application. Premises not explicitly mentioned had to be reconstructed hermeneutically on the basis of further statements in the interview, or they had to be reconstructed rationally. The latter means that an attempt is made to use the principle of charity (“charitable interpretation”) to supplement the missing premises in the way that a rational actor would formulate them in order to provide the most plausible (“best”) justification.
However, we have revised the entire section on the Value Judgment Model in response also to comments from another reviewer (see below). In this revision, we have addressed this particular point as follows:
“When applied to the interviews, descriptive or evaluative premises not explicitly mentioned had to be reconstructed either hermeneutically or rationally. Hermeneutically reconstructed means that the most probable statement containing such a premise was worked out on the basis of further statements in the interview or by interpreting a statement in the context of other statements. Rationally reconstructed means that, using the principle of charity (‘charitable interpretation’), the missing premises are supplemented in the way that a rational actor would formulate them in order to provide the most plausible (‘best’) justification.”
Line 260 “The animal research ethics approach” is used here as if there is a universally recognised concept for how to approach animal research ethics, which is based on a set of principles. I think the principles listed here are a good description of the basic aspects of mainstream animal research ethics, but they are in no way as universally recognised in animal research ethics as the ones by Beuchamp and Childress are in medical ethics.
Authors: Many thanks for this observation. We concede that the original wording may give the wrong impression. We have therefore first changed the following: “Different principlism approaches have been developed for different areas of bioethics. A principlism approach for animal research ethics has been proposed by DeGrazia and Beauchamp [REF]. According to this approach, the suffering of laboratory animals can only be justified […]”. At the end of the passage, we newly included: “Even if there are other approaches to animal research ethics, we considered this approach to be suitable for addressing essential ethical aspects of animal experiments.”
Table 2: WE 1.10-WE1.13 are presented twice.
Authors: Thanks; we deleted it.
Lines 412-414 Unfortunate wording. It’s not bureaucracy (either in the meaning of a governance system where public servants make many decisions or in the meaning that the interviewee refers to, as a complicated system) that is a best practice to ensure validity and integrity. Rather say something like “The review system for animal experiments, which may be perceived as burdensome bureaucracy, is not an end in itself”.
Authors: We agree and changed it as suggested.
Line 447 suggest to revise to “seems to be more obvious in some fields and for some research questions”.
Authors: Thanks for the suggestion, we changed it accordingly.
Lines 480-482 cite Garner et al (2017; see below) who write precisely about the kind and scale of uncertainties are assumed when working with animal models
Authors: We included the reference as suggested.
Reviewer 3 Report
Comments and Suggestions for Authors
Please see attachement

Included in the review
Author Response
Reviewer 3:
As someone with a background of animal research and an interest in non-animal methods and their use (or not use) I was very drawn to the manuscript and the abstract, since the discussion around choices of methods is very lively currently and interests me greatly. However, I found the language very dense, specific and difficult to follow at times and unravelling some of the writing was quite a challenge for me. I think my difficulties are due to the fact that this manuscript is written for an audience which has quite a comprehensive background in philosophy/ethics. However, given that the audience of “Animals” is quite broad with a lot of different professional backgrounds, I assume that there will be other readers like me (with an interest in the topic but maybe not enough background in philosophy to find the reading easy). Hence in general I would advice the authors to re-write/re-phrase the manuscript to make it more accessible and easier to read for a wider interested audience and ensure a better impact.
Authors: Many thanks for the comment. We understand the objection and appreciate that the reviewer is thinking about a better impact of the paper for people outside philosophy/ethics. Unfortunately, we are no longer in a position to completely reformulate the manuscript at this time. However, we have made an effort to simplify sentences in places or to insert additional explanations. We are unable to list these individually in the reply, so please refer to the revised manuscript.
Additional notes on language: Use of italics – The use of italics seems random and is confusing. Sometimes it’s used to indicate an actual technical term (in vitro, Principles of…) and sometimes it’s just random emphasise on words or phrases (for example: whether/how l.45, generations of researchers l.569). The unclear use of italics is confusing and I would strongly advice to use italics only to indicate a specific technical terms and the use it consistently (for example “Value Judgement Model” is sometimes in italics, sometimes not). Use of brackets – Brackets are overused in this manuscript and the use is inconsistent. Sometimes brackets contain alternative meaning, sometimes examples, additional information etc. They severely diminish the flow of reading and often make the authors look indecisive about their writing.
Authors: Thanks for the tip. We have gone through the text and reduced and tried to standardize italics and removed several brackets.
- 12-13: I find this phrasing unfortunate as it somewhat implies that scientists don’t really think critically about which is most appropriate model for their research question, but rather pick a model on the “spur of the moment”.
Authors: Thanks for pointing that out, such an implication was definitely not intended. We refer to the way decision-making works generally in humans, and want to avoid the impression that this is specifically attributed to science. We have therefore rewritten it. It now says: “Moreover, persons often process information unconsciously.”
l.15-17: Either use percentage OR numericals.
Authors: We have adjusted it to only numericals.
l.22 + l. 37: Delete wordcount.
Authors: We deleted the wordcount.
l.24: This is a weird premises. And again seems to imply that scientists don’t think about what are the best models or methods to address their research questions, but seem to pick methods and models at random without thinking about it and have no clue what they’re doing. Which I find bordering insulting. So it would be nice to re-phrase this.
Authors: Thank you for pointing out how this passage can be read. Far be it from us to accuse scientists in the field of animal testing or alternatives of not giving it a second thought! Again, we are concerned with the fact that we humans are generally not always aware (nor can we be aware) of what influences our decisions, and can only improve this to some extent if we make parts of the decision-making process explicit, in ethics especially those parts that involve value judgments. We have rewritten this part of the abstract: “Research model selection decisions in basic and preclinical biomedical research have not yet been the subject of an ethical investigation.”
- 52: This is obviously not the case for veterinary research, which benefits animals and some zoology research, which may benefit the protection of species or the environment. These type of research is not the majority, sure, but especially given that at least one of your interviewee has a veterinary background, I think it should be mentioned here that there is also animal research and testing that benefits animals directly.
Authors: Thanks for the observation, which is important. Three of our interviewees belonged to veterinary medicine. We are, however, from an ethical point of view somewhat critical of the motivation behind animal experiments in veterinary medicine. Of course, it is true that animals may benefit directly from it. However, such research is rarely carried out on the basis of mere altruism towards animals, so to speak, but on the basis of human interests in having healthy animals (in food production, in zoos, as pets etc.). We therefore changed the text to the following: “This is only different in the comparatively much smaller research in veterinary medicine, which is intended to benefit animals; however, it can be critically argued that also this animal research is ultimately only carried out in order to further human interests (e.g. reducing economic losses in livestock farming due to diseases, or the treatment of pets).”
L.56: I think there is not enough information what “sufficient scientific value” means in this context to be a stand-alone. As a stand-alone phrase it implies that animal experiments lack scientific merit in general, while the authors use the phrase in regards to very specific aspects of reproducibility which may have an impact on experimental results.
Authors: We changed it as follows; but please consider that the whole section that entailed this sentence has been revised: “Although such epistemological questions can also be addressed independently of ethical considerations, it is also argued that animal experiments which do not have sufficient scientific value fail to be ethically acceptable even if all other ethical requirements are fulfilled; Strech and Dirnagl, for example, propose three principles to safeguard and enhance the scientific value of animal research: robustness, registration and reporting [REF].” Further details can be found in the reference given: 3Rs missing: animal research without scientific value is unethical - PMC (nih.gov)
- 67-72: This neglects the obvious fourth option, to do both – animal-based and animal-free methods. Which is what in reality most researchers do.
Authors: That is true. We added: “In this context, 'making choices' refers to the many small decisions about how to approach a particular research question, or even just one aspect of it. Countless such choices can be made in research projects, in working groups, and even more so in a researcher’s career. As researchers usually pursue many research questions at the same time – have different experiments running –, they often work with animal and non-animal models in parallel [REF].”
- 111 (and other): Unless this is indeed specific about research in regards to diseases, I don’t think “disease model” is a good term to use, since neither all animal models nor all nonanimal models are disease models. Maybe “research model” may be a better term.
Authors: We actually used “research model” in an early version of the manuscript, but switched to “disease model” after consulting someone in animal research. However, we find the argument here convincing and have switched back to “research model”. Only in two places have we left “disease model” as an example of what can be meant with “research model”.
l.112 ff.: Provides the theoretical and methodological background for what?
Authors: For the analysis and evaluation of decision-making, esp. of course in this context regarding the choice between animal experiments and alternative methods. We changed the whole sentence also as reaction to a comment of reviewer 2:
“In this context, ethics, as the systematic exploration of values, norms and principles, and as the critical examination of arguments involving (moral) value judgments, provides a relevant theoretical and methodological background for analysing and evaluating decision-making – which does not preclude other relevant approaches, such as those from (therio)epistemology or cognitive psychology.”
- 156-167: I don’t think it’s necessary to go into such details about which researchers were not interviewed.
Authors: We agree that is was addressed to prominently and put most of the corresponding part in the limitations section. It there now says: “We faced several challenges in the recruitment phase. A first attempt, to limit and define the scope of the decisions-making-situation, focused on Alzheimer’s research. We contacted a large number of researchers in this field multiple times by mail and telephone. We only received one response. One possible reason for the failed recruiting could have been the COVID-19 pandemic. The interviews were planned and conduced in the period from April to June 2020 (during lockdown in Germany). It is possible that this circumstance led to a lack of availability. However, we were unable to verify this hypothesis. We then decided to change our strategy and no longer limit ourselves to Alzheimer’s research.”
- 178-179: Is this standard procedure or best-practice, to stop recruiting when you think you have enough data and not decide beforehand how much data you will collect?
Authors: This is indeed a proper way of determining the end of the recruitment. Further findings in the data (in this case, further reasons not yet included in the spectrum) cannot be ruled out, but the gain in knowledge is “diminished” and the effort becomes much more tedious. Specifically, perhaps 10 more interviews would have to be conducted to identify one new aspect. We explain this in a new paragraph in the limitations. It now says:
“The theoretical saturation in this project was reached; nevertheless, it is important to note that we might have missed reasons that could, for example, by using of different ethical theories or approaches. Regarding the latter, we have not analysed or discussed (therio)epistemological issues, and have decidedly not taken a perspective from the philosophy of science, as one of our colleagues in R2N did [REF]. Our focus was on the ethical value judgments and associated reasons. However, we have implicitly addressed epistemological reasons insofar as they are presupposed in certain ethical norms. For example, in norms that demand scientific validity; but basically already in the Principle of No Alternatives, which presupposes the answer to the epistemological questions of what knowledge can be generated by animal experiments or alternative methods and how good this knowledge is. Furthermore, other reasons might have been found in a sample with more female respondents. In our sample, senior male researchers in Germany where the most dominant group. The spectrum presented can and should therefore be complemented by further qualitative research. However, it is important to notice that qualitative research strives for diversity and range of topics and not representativeness in a statistical sense. On a personal level, not all 66 reasons are likely to be equally important, and some will not resonate at all. Our qualitative approach does not allow us to say anything about the frequency of the reasons given. It is possible that some reasons are primarily due to particularities such as the bureaucratic effort and associated delays in Germany or in the EU. These reasons may be less relevant for people working in other contexts, but this is not a limitation in qualitative research.“
l.226 ff.: In general I find the explanation of the “Value Judgement Model” very difficult to follow. It’s a model which is quite specifically based in philosophy/empirical ethics and needs some background knowledge in philosophical/ethical concepts and theories to understand. As mentioned before, given that the audience of “Animals” can be very mixed in regards of their professional background, it will probably be advisable to explain this in more detail and a language, which is more accessible for a broader audience. The cited literature ([19]) does explain the model in detail, but it is clearly aimed towards an audience, which has a background in philosophy and hence that paper is also not easily accessible for a reader without that background knowledge. I would therefore advise to give additional explanations about the model (maybe in supplementary materials), targeted specifically to a broader audience.
Authors: (As there were several comments on the Value Judgment Model (VJM) and its application, we try to cover most of it in the response here and often just refer to this response later).
Thank you for your feedback. Actually, one reason why we did not want to explain the VJM in too much detail and understood it more as a necessary explanation of the methodological background was to keep it simpler for non-philosophers/ethicists. This is another reason why we decided to focus on the reasons themselves in the presentation of the results and only give examples of the application of the model to some of the reasons, and not in an over-formal way (see the newly inserted Table 1 regarding the more formal representation). The manuscript would have been even longer if we had reproduced the application in full for each example. So, the application was part of what we were doing – which is why it is mentioned in the methods section – but should not be the focus of this article.
We maintain that we cannot present the – explicit – results of the application for every example discussed because it would be too extensive and because, as we said, we want to focus on the reasons as the central result. However, we have of course tried to explain the VJM somewhat better and what role it played in the coding as well as in the analysis. To this end, we have added the new Table 1 mentioned above, which uses three examples to illustrate what a complete analysis using the VJM can look like. We have also referred additionally to the respective evaluative/descriptive premises in selected examples where we did not do so earlier. Furthermore, in the new Table 3 (old Table 2) we have added whether the reason is an evaluative or descriptive assumption.
We hope that the improved explanation, the illustration in the new Table 1, the additional explanations in individual examples and the additional information in the new Table 3 will make the VJM and its application a little easier to understand.
- 247-249 - I have the most problems to understand the practical application of the model and why exactly it is helpful and needed. Basically I don’t understand which role the model plays to get from an interview to the results in l. 321-327 & table 2. How was the model used to get the results in table 2 and why was it necessary to use? Would you not have been able to filter out the statements in table 2 without the model? I’m really trying to understand the advantage and necessity of the model, but I have real difficulties to see how and why it was used. Therefore I would really appreciate if the authors could explain with an actual example how the model is used when coding the interviews and compiling the results (again, maybe in some supplementary materials).
Authors: See comment above. However, we would also like to point out that this is not an educational publication that teaches readers how to code qualitatively. Such a demonstration would be made in textbooks, but not in a research article that is ultimately designed to present results. We have therefore tried to describe a bit more detail why such a model is important for coding, but have refrained from providing a step-by-step illustration, so to speak.
Table 1 – It would be good to also give an overview of the professional background of the interviewed experts.
Authors: Thank you for the suggestion. We have added some additional information in the table (now table 2).
- 313 – Shouldn’t it be “some boundaries” instead of “any boundaries”?
Authors: We changed the wording to: “However, it is clear that there does not seem to be a clear boundary as to what is considered an alternative and what is not.”
Table 2 – WE1.10 – WE1.13 is in there twice
Authors: Thanks, corrected.
l.353-354 – Too speculative. There is absolutely no proof whatsoever to “imply a reference to a general attitude to “delegate””. This is not interpretation, but pure speculative guesswork. Delete or re-phrase.
Authors: Thanks for the remark. We agree that this is not evident enough from the paraphrase alone. However, it was raised in the interviews, which is why we do not want to completely dispense with the statement. We have rephrased the majority of the section, also in connection with the desire to refer more systematically to the principles in the “ethical dimensions”:
“In the category of personal attitudes, there is, not surprisingly, a certain range of reasons. Not all reasons in this category refer only to internal or subjective criteria – as is the case when, for example, emotions play a stronger role (evaluative premise): “I choose the alternatives because experiments with animals are stressing me emotionally/psychologically” (PA1.3, Tab. 3). For instance, the reason (evaluative premise) “I choose the animal model because it can be justified and is ethically acceptable” (PA2.1) also refers to an external basis for justification, even if it is not further explicated in the reason itself. The evaluative character of this reasoning becomes clear, as the reference that animal experimentation is justified and ethically permitted could be based on the current societal norm setting, which basically allows animal experiments. One could also refer to the defined procedure for third party approval (competent authorities), which can be regarded as safeguards for ethically defensible research involving animals. However, the reason could also refer to scientific and ethical arguments that are to be understood independently of or in addition to an examination by an animal ethics committee, but which initially elude further intersubjective examination due to a lack of explication – it is, as an unspecified reason, initially only an assertion that the animal experiment is justified, without reference to ethical principles. So, personal attitudes can be grounded in a) values/principles (“protecting a human being from ineffective or harmful drugs is a higher value than refraining from animal experiments,” PA2.4), b) emotions (PA1.3 see above) or c) interests (“I am curious to try new things,”, PA1.1) – all evaluative premises.
Ethical dimensions: None of these personal attitudes is ‘prima facie’ ethically better or worse. However, the background of the evaluative premises can be further explored. The reason of being curious (PA1.1), for example, would not be ethically sufficient as the sole reason for refraining from animal experimentation in view of the Principle of Expected Net Benefit, if the use of an alternative would reduce the expected social benefit. Furthermore, a general attitude to ‘delegate’ the decision to a ‘higher’ level of decision-making could be problematized, as there is then no ethical reflection and justification of one’s own. The reasons discussed also contain descriptive statements that can be analysed. Following the VJM, the descriptive premise “the established approval procedure is most likely to lead to a ‘correct’ decision about moral permissibility” could be rationally reconstructed from the interviews, as this makes the delegation of ethical assessment to e.g. animal ethics committees plausible. In the case of independent reasons, however, the “justified” could also refer to scientific or epistemological reasons related to the fulfilment of the norm of scientific validity.”
l.362 – What does “rationally reconstructed” means exactly?
Authors: We explain it now (together with hermeneutically reconstructed): “When applied to the interviews, descriptive or evaluative premises not explicitly mentioned had to be reconstructed either hermeneutically or rationally. Hermeneutically reconstructed means that the most probable statement containing such a premise was worked out on the basis of further statements in the interview or by interpreting a statement in the context of other statements. Rationally reconstructed means that, using the principle of charity (‘charitable interpretation’), the missing premises are supplemented in the way that a rational actor would formulate them in order to provide the most plausible (‘best’) justification.”
l.375-376 – Please give a concrete example for primary and especially secondary interests.
Authors: We re-phrased the sentence. It now says:
“These considerations are referred to as secondary interests in the concept of conflicts of interest: interests that are not directly related to professional activities as researcher (e.g. personal curiosity of a scientist, completing a doctoral thesis as part of pursuing a career, or just earning a living). It is important to note that primary and secondary interests are often not in conflict with each other, but can even have positive effects. However, if they are conflicting with primary interests (e.g., the pursuit to produce relevant and valid findings, disseminating results in the scientific community or adherence to the six principles of animal research ethics), they should not influence the professional judgement inappropriately [REF].”
- 386 – What does “fictitious” mean here? Isn’t PA1.2 an example for media attention?
Authors: We agree and deleted “fictitious”.
l.401 ff. – The authors seem to assume that the aim of the bureaucratic process when applying for a project permit is to ensure scientific validity of a project and hence that the authorities have the final say on whether a project is scientifically valid or not (l. 412-414). This is not the case. This bureaucratic process is not in place to judge the science, but ensure best animal welfare for the animals in a project. Authorities shall verify if a project is scientifically justified (see EU-Directive 2010/63). They may critique permit applications in regards to the science but it is not within their authority to make the final judgement about the science of a project. This final judgement and responsibility lays with the researcher (protected also in Article 13 of the EU Charter of fundamental rights). I know this seems like a trivial and minor distinction and in practice, authorities do give a lot of input on the science of a scientific project when researchers apply for a permit. But it’s an important distinction to note that authorities do not hold power over making final decisions on science since this would violate the freedom of science. I’m assuming the interviewed are all based in Germany. The bureaucratic process for acquiring a permit in Germany is famously long and regularly exceed legally implemented deadlines (see Report on the implementation of EU-Directive 2010/63 from 2020). Furthermore, the German application process contains exceeding amounts of paperwork. Both these facts make the bureaucratic process burdensome and bring huge insecurities for the researcher, which can summate to a real hindrance of research. I really doubt it’s about efficiency of research, but actual hindrance of conducting research. But Germany is a special case in regards to lengthy bureaucratic processes (as far as I can see it), so I’m not sure if the view of German researchers on bureaucracy and how it influences their choice on methods is generalizable per se. Hence I would reconsider this paragraph or at least add some footnote on the special case of German bureaucracy.
Authors: Thank you very much for the good considerations. We agree that it is not the competent authorities or animal ethics committees that ultimately decide on the science; that is not their job. It is true, however, that we would say that part of the review process should check whether the animal experimentation project is sufficiently scientifically sound (the reviewer also seems to agree with this, if we understood her/him right).
However, we actually were more concerned with the following: If someone chooses an alternative method because it is bureaucratically simpler (“more efficient”), but an animal model would be scientifically more meaningful (can produce more valid results), it would be ethically problematic to choose the alternative research model. The choice should therefore not be made solely on the basis of less bureaucracy. However, this does not seem to have been made clear enough in the text, so reworded the passage:
“Some scientists argue that “alternatives are associated with smaller amounts of lengthy bureaucracy (e.g., no approval procedure)” (WE1.2). Ethical dimensions: The evaluative assumption behind this might be that “lengthy bureaucratic processes are bad for / are a hindrance to research.” It should be noted that this is a specific perspective that emphasizes efficiency, which can be understood as a shared value or (ethical) principle in science, and especially in research within the health-care system, given that public resources are being spent. However, in the research context, scientific validity or ethical integrity is probably more important than efficiency, i.e., a higher value must be placed on validity or integrity when comparing research models. This does not mean that efficiency cannot be taken into account, but decisions should not be made only in favour of efficiency – choosing an alternative method only because it means less bureaucratic effort would be ethically questionable when scientifically, an animal model would be preferable, as this also may violate the Principle of Expected Net Benefit (= social benefit is only possible when the research model is meaningful for the research question and the results robust). Additionally, the review system for animal experiments, which may be perceived as burdensome bureaucracy, is not an end in itself, but is, in a certain way, an operationalization of the Principle of Sufficient Value to Justify Harm and the Principle of No Unnecessary Harm.”
Furthermore, we report the origin of the interviewees in table 2 and addressed that issue (German context) in the limitations.
l.435 – Which ethical principle specifically?
Authors: We re-phrased the sentence, as at this point, we do not refer to a specific principle: “Similar to bureaucratic requirements, actions based on existing or required infrastructure are not per se unethical”.
- 457-460 – I find this unclear. What exactly does this mean? What’s the conclusion?
Authors: Thanks for the critical question. We agree that the point was not clear enough and have revised it and added another ethical consideration:
“This reason is similar to the earlier example, in which an alternative research model is only chosen because of the possible lower bureaucracy, although animal models would perhaps make more scientific sense and thus make a higher social benefit possible (Principle of Expected Net Benefit). In the example here, a problematic influence by secondary interests (number of publications as a career benchmark) is perhaps even more conceivable. However, if it is assumed that the results can be used in a comparable way by applying the alternative methods, the tide can turn. Only published findings can fulfil the central promise of generating value through research, as only what is published can be taken up scientifically and, at best, later translated into health care. If usable results can be produced and published more quickly by using alternative methods, the value of animal experiments for a comparable question is reduced, as this also calls into question the fulfillment of the Principle of Sufficient Value to Justify Harm: The sufficient expected social benefit to inflict harm on animals is reduced if sufficient value can be generated via alternative methods, and then even faster.”
l.462 – Pragmatic considerations such as?
Authors: We have inserted three examples here: “In contrast to such examples of reasons that can be attributed more or less directly to justifications using ethical principles, there are pragmatic considerations, for example, that “there is suitable infrastructure on site” (WE 1.4), that something could be ”realized more quickly” (WE 1.3) or that “there are also legal requirements” (S2.9)”.
l.463-466 – The whole sentence sounds like it’s missing a “but…” and another sentence.
Authors: We re-phrased the passage:
“The different dimensions of use or benefit should be considered here. For research to be of benefit to society, the results must be published in full and in a timely manner. There is also the legitimate interest of the individual researcher in commercialisation by industry (“industry often demands results from animal models, otherwise it is hardly possible to commercialise our results”, S2.6). Even if commercialisation appears to be necessary in the current scientific system to continue research through better funding, the Principle of Expected Net Benefit should be carefully respected. So, such secondary interest should be given less weight in the event of a conflict, as the primary interest should be the aim of rapid and comprehensive publication.“
l.466-478 – Wouldn’t that belong to “work environment”? Especially since the example given is from work environment.
Authors: We agree and have moved it accordingly.
L483-485 – I think the sentence is missing some verb at the end. (“…that the common and known uncertainties in animal models (missing verb).”)
Authors: In fact, no verb is missing, but there is a spelling mistake: it was meant “than”, not “that”; we also changed the wording slightly: “Perhaps these uncertainties related to the use of alternative methods impress more than the common and known uncertainties in animal models”.
- 497-502 – I don’t understand how these sentences connect to the first part of the paragraph. The first paragraph cites two reasons (WE1.10/WE1.13), both in favour for using non-animal methods (because of funding). But the sentences l. 497-502 now relate to a choice pro animal method because of funding? I don’t understand what/how the authors argue here.
Authors: As already acknowledged in the similar comment by reviewer 2, the direct reference was missing here; the ethical discussion was more about the question of how it should be assessed if these reasons did not exist. We have now revised this completely:
“Researchers have to continually find sources of funding and, thereby, orient themselves to the external (public) research funders. Investments in promising in vitro or in silico methods (such as organoids, organ-on-chip, human-cell-based models or computer simulations), or more and better systematic reviews as desk research, as well as the qualification of personnel to use them, is a strong driver for a shift more towards alternative methods, and if the investment costs are funded, this is beneficial (WE1.10+13). Ethical dimensions: Ethically, it is rather trivial that if funding is available for the use or development of alternatives, the researchers are competent to use or develop them and it can be assumed that comparable research questions with comparable (expected) social benefit can be addressed with them, the alternatives are preferable to animal experiments. If a viable alternative is obviously available or can probably be developed, an animal experiment would violate the Principle of No Alternative, which is quite uncontroversial and also, as mentioned in the introduction, legally defended. It is therefore more interesting to ask what happens if the funding for alternatives is not available or is insufficient, even though it would be conceivable to use or, especially, develop a viable alternative. As the resources are allocated in different funding lines, it may be the case that no funding is available for the development of infrastructure and the qualification of staff for research with alternative methods, but funding is available for the acquisition of a certain number of mice. Formally, an evaluative premise such as “It is good/right to conduct an animal experiment (at least better than not conducting any research) if an alternative is generally available but cannot be adequately funded in this case” is able to provide a justification. However, accepting such a premise obviously opens up countless exceptions to the Principle of No Alternative Method. Thus, if alternatives would be available or could conceivably be developed, insufficient funding is not an ethically convincing argument for the use of animal models. Still, with reference to the so-called ought-implies-can principle – simplified: one can only demand normatively what is also realistically realizable (e.g., [REF]), it could be argued that animals are the only possible research model in this case. Nevertheless, following the Principle of No Alternative Method, one could still argue that the conclusion in these cases should rather be “no animal experiment if an alternative is available but financially unfeasible” – even though this must be examined in each individual case (e.g., to what extent is the alternative available but not feasible, to what extent would it be suitable).”
l.518 – How do we know the reasons referring to descriptive assumptions?
Authors: In the new Table 3 (old Table 2), we have now added in brackets at the end of the stated reason whether the assumptions are considered evaluative or descriptive.
l.522 – What is meant by “it”?
Authors: We deleted “it” and changed the sentence to: “[…] such reasons should be guiding considerations in decisions”.
- 524 – What are some of the “pragmatic contextual factors” and why?
Authors: We added: “(e.g. infrastructure or funding)” (however, complete section has been revised, see next comment).
l.523 – 533 – Can’t quite follow how this relates to the first part of the paragraph. Maybe rephrase for clarity.
Authors: As we said in our response to reviewer 2, we acknowledge that this section can be improved. It now reads as follows:
“Reasons such as “I want to avoid animal suffering” (A1.1) are very straightforward ethically, as they can be directly subsumed under the Principle of No Unnecessary Harm or to more general ethical norms that are oriented towards non-maleficence. If this kind of reason is understood in absolute terms, it inevitably leads to an abolitionist position, i.e. the complete abolition of animal experiments; this would go beyond the Principle of No Unnecessary Harm. However, if it is understood in relative terms, it leads to the well-known question of weighing up what kind and what degree of harm is permissible (“necessary”) in view of a possible benefit, which is represented in the Principle of Sufficient Value to Justify Harm. Defined constraints for weighing are also conceivable in favour of scientific considerations (e.g., validity) and their related ethical values (e.g., expected social benefit). Still, the appropriateness of the weighing should always be checked in a review process by third parties to mitigate possible biasing effects of secondary interests (see Personal Attitudes).
Other interviewees pointed out the consequences of failure of the experiments and compared this between animal experiments and alternative experiments: “The consequences of a failed experiment are much less critical than in animal experiments” (A1.3). Ethical dimensions: This reason, besides following in a way the Principle of No Unnecessary Harm – as unnecessary harm would be the case when the animal experiments fail –, seems also to refer to a form of the precautionary principle (“err on the side of caution”): It is better to lose a potential social benefit that could have been generated by animal experiments than to cause possible unnecessary harm to animals. When balancing the principles, the Principle of Expected Net Benefit is therefore given less weight than the Principle of No Unnecessary Harm.
The reason A2.1 for conducting animal experiments is somewhat more complex: animal experiments “[…] are necessary and before someone does it who doesn’t care about animals, I prefer to do it myself”. Here, “objective” assessments – the experiment is scientifically necessary – are combined with “subjective” attitudes. Ethical dimensions: On the one hand, it is clear that the Principle of No Alternative Method must be fulfilled; otherwise, the animal experiment would not be necessary. On the other hand, however, the interviewee has doubts that the Principle of No Unnecessary Harm and perhaps also the Principle of Upper Limits to Harm are observed by all animal researchers, which is a concern for him/her. This means that even if he/she might prefer not to carry out animal experiments, he/she will still carry them out to ensure that at least in these animal experiments the principles are met.
Finally, when considering the reasons for the Work Environment and Personal Attitudes categories, it is worth reflecting on the extent to which it is permissible to restrict animal welfare in favor of mere pragmatic contextual factors (e.g. infrastructure or funding). Ethical dimensions: When an ethical stance is taken that always gives priority to the moral point of view or the moral position over other points of view, especially over self-interest (e.g., [42, 43]), the answer is relatively clear: it is never permissible – although perhaps not always avoidable in reality. In any case, it then only seems to be defensible at all if the pragmatic reasons can (indirectly) refer back to ethical values or principles, e.g. to the Principle of Expected Net Benefit. In a sense, the argument would then be that not conducting animal experiments or conducting them in a more limited way would lead to fewer expected social benefit, which is why at least some pragmatic reasons in a non-ideal world must be regarded as justification. However, the extent to which such an argument can be used in individual cases must be subject to critical discussion.”
l.347-533 – In general I think the “ethical dimensions” need to refer more systematically to the ethical frameworks laid down in
Authors: We have added more explicit references to the principles in various places. However, we still do not do this “mechanically” for every discussion point. That would simply become too long (the reviewer may already see how much the word count has increased as a result of all the additions). We have only ever talked about an exemplary discussion.
l.259-275. For every ethical dimension it should be stated which “principle” is relevant here.
Authors: See response above.
l.608-609 states that for the ethical analysis the reasons were reconstructed into relevant descriptive and evaluating premises; so it would be good to actually name the relevant premises specifically for each ethical dimensions. This would make it easier to follow and see the impact and necessity of the Value Judgement Model for the ethical analysis.
Authors: Here, too, it is not possible to do this everywhere without making the text many times longer. The explication of the premises was also only intended as exemplary for the manuscript. By explaining the VJM in more detail, we hope to have made those passages in which we explicitly refer to the descriptive or evaluative premises more comprehensible.
- 540 – Which are the well-known issues? Some examples would be helpful.
Authors: We added: “(e.g. animal welfare or harm-benefit analysis).”
l.569 – I’d use a different, more specific term instead of the generic “new technology” because there is always some sort of new technology in the laboratory, so every researcher has worked with new technology at some point.
Authors: We changed it to: “Investments in promising in vitro or in silico methods (such as organoids, organ-on-chip, human-cell-based models or computer simulations), or more and better systematic reviews as desk research, as well as the qualification of personnel to use them, is a strong driver for a shift more towards alternative methods, and if the investment costs are funded, this is beneficial (WE1.10+13).“
- 578 – Which are the “fundamental value questions”?
Authors: We elaborate more and it now says: “What are meaningful outcome dimensions for making statements about effects in humans, what implications arise from the interpretation of the data, and how should be dealt with uncertainty and risk?“.
l.580-581 – This must be some specific jargon used only in philosophy because I would say it is not necessary to apply qualitative analysis to observe reasons. I’m 100% certain I have heard all the reasons in table 2 before in conversations or discussions with colleagues and I didn’t apply any analysis. So, to prevent other readers without a philosophy background to be as confused as I am about this sentence, it would probably be good to re-phrase this.
Authors: Generally spoken, the more the concept of “reason” is understood as a premise that is supposed to be logically validly linked to a conclusion, the more reconstructive elements come into play, unless – as is often the case in philosophical texts – the most important premises are explicitly presented and the function they fulfill in an argument’s structure is explained. This, however, is not the rule in everyday life, incl. ethical everyday reasoning. That was the background to the claim.
However, we agree that the sentence was poorly formulated; actually, we should rather have talked about “value judgments” (understood as results of specific arguments) that are not easily “observed”, but have (in part) to be reconstructed, not the reasons.
But we have decided to formulate it completely differently. We deleted the first sentence and added the following instead: “We are often not fully aware of all the considerations that lead to decisions. Moreover, it is hardly possible to describe the full process of consideration within an explanatory statement. By paraphrasing and reconstructing, it was possible to create a spectrum of 66 distinct reasons, each based on (several) statements from the interviews.“
- 585-587 – As mentioned above, I can’t follow the practical application of the model and the impacts it’s having on the coding. Therefore, I can’t follow this statement since I can’t really see the evaluative and descriptive premises. (same for l.608-609)
Authors: See earlier comments regarding the VJM and its application.
- 595-598 – Please clarify.
Authors: We have added some clarification:
“[…] Being clear about this, and especially trying to make the evaluative premises more explicit, can help to check whether the value judgment is sufficiently supported. Because if it is assumed that a value judgment is the conclusion of an argument, it is not sufficient to merely demonstrate the truth of the descriptive premises. It must also be shown why the evaluative premises are justified, e.g. by referring back to established or at least intersubjective defensible ethical principles or values.”
l.630-632 – Quite honestly speaking, in order to get sufficient compliance and understanding by the researchers for these measures, the measures must not increase additional workload or paperwork for researchers. Since, as the authors stated, decisions may be predetermined and the researchers may lack of understanding as to why they need to add more workload and may see it as simply “jumping through hoops” without much benefit to practical work. Suggestion for additional literature: The British “Biotechnology and biological sciences research council” published a survey in 2022 (“on the use of models in research”), which specifically looked into the choice of methods. Maybe this might be interesting for authors to include in their manuscript.
Authors: We thank you for the hint. We added: “however, they must not lead to more paperwork being required and make practical work more difficult.” We also reference the survey.
Reviewer 4 Report
Comments and Suggestions for Authors
I review this paper as a Ph.D. level bioethics professor who has served on an Animal Care and Use Committee to review and approve undergraduate research projects using animals and who has done some undergraduate teaching on research ethics pertaining to the use of animals. From this perspective, I have nothing but praise for this paper as an addition to the published literature. Among other audiences, I foresee it being used as a pedagogical tool by academics. As an empirical study of the reasons which determine researchers to use animals or an alternative, this paper can be used to show students what is in fact going on in the real world. The empirical study, combined with the authors' ethical analysis, can be used to stimulate students to think about and critically assess their own reasons for wanting to use (or to refrain from using) animals in their own research projects.
The introductory section provides solid motivation for the study. The results are meticulously presented and analyzed in detail from an ethical point of view. The authors are conversant with the published literature on research ethics pertaining to animals. Overall, this paper is one of the most substantive studies on this topic that I have encountered.
Author Response
Reviewer 4:
I review this paper as a Ph.D. level bioethics professor who has served on an Animal Care and Use Committee to review and approve undergraduate research projects using animals and who has done some undergraduate teaching on research ethics pertaining to the use of animals. From this perspective, I have nothing but praise for this paper as an addition to the published literature. Among other audiences, I foresee it being used as a pedagogical tool by academics. As an empirical study of the reasons which determine researchers to use animals or an alternative, this paper can be used to show students what is in fact going on in the real world. The empirical study, combined with the authors' ethical analysis, can be used to stimulate students to think about and critically assess their own reasons for wanting to use (or to refrain from using) animals in their own research projects.
The introductory section provides solid motivation for the study. The results are meticulously presented and analyzed in detail from an ethical point of view. The authors are conversant with the published literature on research ethics pertaining to animals. Overall, this paper is one of the most substantive studies on this topic that I have encountered.
Authors: Thank you very much for your kind and encouraging words!
Round 2
Reviewer 1 Report
Comments and Suggestions for Authors
No further comments
Reviewer 2 Report
Comments and Suggestions for Authors
Congratulations to a very valuable piece of work! No further suggestions.
Reviewer 3 Report
Comments and Suggestions for Authors
I appreciate the extensive additional work the authors invested into the manuscript. I especially appreciate some of the more accessible language in the new edits. That was very helpful for me and I think it will make it easier for more readers to follow the interesting and valuable work of the authors.
Thank you!